# New hominin remains and revised context from the earliest *Homo erectus* locality in East Turkana, Kenya

Ashley S. Hammond [1,2✉], Silindokuhle S. Mavuso[3], Maryse Biernat[4], David R. Braun[5,6], Zubair Jinnah [3], Sharon Kuo [7], Sahleselasie Melaku[8,9], Sylvia N. Wemanya[10,11], Emmanuel K. Ndiema[10], David B. Patterson[12], Kevin T. Uno [13] & Dan V. Palcu [14,15]

The KNM-ER 2598 occipital is among the oldest fossils attributed to *Homo erectus* but questions have been raised about whether it may derive from a younger horizon. Here we report on efforts to relocate the KNM-ER 2598 locality and investigate its paleontological and geological context. Although located in a different East Turkana collection area (Area 13) than initially reported, the locality is stratigraphically positioned below the KBS Tuff and the outcrops show no evidence of deflation of a younger unit, supporting an age of >1.855 Ma. Newly recovered faunal material consists primarily of $C_4$ grazers, further confirmed by enamel isotope data. A hominin proximal 3rd metatarsal and partial ilium were discovered <50 m from the reconstructed location where KNM-ER 2598 was originally found but these cannot be associated directly with the occipital. The postcrania are consistent with fossil *Homo* and may represent the earliest postcrania attributable to *Homo erectus*.

[1] Division of Anthropology, American Museum of Natural History, New York, NY, USA. [2] New York Consortium of Evolutionary Primatology (NYCEP), New York, NY, USA. [3] School of Geosciences, University of the Witwatersrand, Johannesburg, South Africa. [4] School of Human Evolution and Social Change, Arizona State University, Tempe, AZ, USA. [5] Department of Anthropology and Center for Advanced Study of Human Paleobiology, The George Washington University, Washington, DC, USA. [6] Department of Human Evolution, Max Planck Institute for Evolutionary Anthropology, Leipzig, Germany. [7] Department of Anthropology, Pennsylvania State University, University Park, PA, USA. [8] Authority for Research and Conservation of Cultural Heritage (ARCCH), National Museum of Ethiopia, Addis Ababa, Ethiopia. [9] Paleoanthropology and Paleoenvironment Program, Addis Ababa University, Addis Ababa, Ethiopia. [10] Archaeology Section, Department of Earth Sciences, National Museums of Kenya, Nairobi, Kenya. [11] Department of Archaeology and History, University of Nairobi, Nairobi, Kenya. [12] Department of Biology, University of North Georgia, Dahlonega, GA, USA. [13] Division of Biology and Paleo Environment, Lamont-Doherty Earth Observatory of Columbia University, Palisades, NY, USA. [14] Paleomagnetic Laboratory 'Fort Hoofddijk', Utrecht University, Utrecht, Netherlands. [15] Instituto Oceanográfico, Universidade de São Paulo, São Paulo, Brazil. ✉email: ahammond@amnh.org

The KNM-ER 2598 specimen from East Turkana, Kenya is widely recognized as significant because it is one of the oldest fossils attributed to *Homo erectus*[1–5]. KNM-ER 2598 is a thick hominin cranial fragment preserving much of the central occipital bone, including portions of the lambdoidal suture and a distinctive *Homo erectus*-like occipital torus (Fig. 1)[6]. This fossil was collected from the outcrop surface in 1974 and was initially reported as originating from approximately the level of the KBS Tuff in East Turkana collection Area 15[6]. Later work would refine the stratigraphic placement of KNM-ER 2598 to about 4 m below the KBS Tuff and interpret the age as 1.88–1.9 million years (Ma) ago[1,7].

If KNM-ER 2598 is dated to nearly 1.9 Ma, it is the second chronologically oldest specimen with morphological affinities to *H. erectus*. The DNH 134 neurocranium from Drimolen is the oldest known *Homo erectus* specimen[8]. DNH 134 is a juvenile individual, which makes a categorical species attribution difficult to establish, but it has features (e.g., a teardrop-shaped superior profile, flat squamosal suture) which strongly favor a *H. erectus* attribution[8]. DNH 134 was recovered from a deposit with reversed paleomagnetic polarity with an associated uranium-series electron spin resonance (ESR) date of 2.04 Ma[8], indicating that this specimen was deposited sometime within the C2r.1r reversed subchron. The most recent Geomagnetic Polarity Time Scale (GPTS 2020) associate the C2r.1r chron to the 1.934–2.120 Ma time interval[9,10]. Both DNH 134 and KNM-ER 2598 are critically important fossils because they are slightly older than those recovered from Dmanisi in the Republic of Georgia. The Dmanisi hominin fossils may be as old as 1.78 Ma[11], and occupation of the site appears to extend to 1.85 Ma[11]. The Dmanisi dates, which approach the dates for the African *H. erectus* specimens, raise the possibility that *Homo erectus* origins could have occurred in Eurasia rather than on the African continent[4].

Accordingly, KNM-ER 2598 is key to anchoring the earliest evolution and dispersals of *Homo erectus*, but some authors have raised doubts about the age of KNM-ER 2598[12,13]. It has been suggested that the altimetric position of KNM-ER 2598 below the KBS Tuff may have resulted from deflation of a stratigraphically younger horizon (e.g., KBS Member) that is no longer visible on the exposed upper Burgi Member outcrop surface[12,13]. Regrettably, few details regarding the provenience of the fossil were offered in the initial publications[6,14]. Given the importance of KNM-ER 2598 for placing the early evolution and dispersal of *H. erectus* within Africa, it has become essential to provide a geochronological context for the KNM-ER 2598 locality.

Here we report on new investigations into the KNM-ER 2598 site location, geology, and paleoecology. The geological data presented here support the interpretation that the KNM-ER 2598 occipital derives from the upper Burgi Member of the Koobi Fora Formation, conservatively dating the fossil to >1.855 Ma. Our findings also correct the location of the KNM-ER 2598 locality, demonstrating that it is situated within the boundaries of collection Area 13 rather than Area 15 in East Turkana. We report on a new hominin ilium and metatarsal recovered within close proximity of where KNM-ER 2598 originated. These hominin fossils are consistent with *Homo erectus*, potentially making these the oldest postcranial fossils attributable to the taxon. Finally, we contextualize the paleohabitat in East Turkana Area 13 through faunal abundance data, isotopic analyses of mammalian dental enamel, and petrographic data.

## Results

**Identification of the locality**. We used field-based reconnaissance combined with historical imagery to identify the KNM-ER 2598 locality (Supplementary Figs. 1–3). We used Google Earth imagery to approximate the geospatial location of KNM-ER 2598 in geographic coordinates from historical aerial photographic records housed at the National Museums of Kenya (Supplementary Fig. 1). The aerial imagery from the 1970s document that KNM-ER 2598 was actually discovered within the boundaries of collection Area 13. Photographs from the 1974–1975 field seasons confirm that collections were taking place in collection Area 13, based on landscape features that are still identifiable (Supplementary Fig. 2). Upon physical inspection of the reconstructed location, a large collapsed sandstone cairn was identified at approximately the same coordinates (N 4.26984, E 36.33848, WGS84; altitude 413.5 m above sea level) reconstructed from the aerial imagery. The sandstone cairn (Supplementary Fig. 3) is consistent with the markers used in the 1970s, prior to the positioning of cement plinths as hominin markers.

**Geological context and age**. Survey of the rediscovered KNM-ER 2598 locality and nearby areas within Area 13 allowed us to identify two distinct aggregates (Burgi and KBS Members) and a tuff horizon (the KBS Tuff) that marks the boundary between the two units (Fig. 2). The tuff horizon is missing in some locations due to erosion episodes in the KBS succession. However, the eastward interpolation of the geologic data for this ash level and topography coincides with coordinates for a known KBS Tuff sample (IL02-128 from ref. [15]; Fig. 3c). The distinct sandstones associated with the KBS and upper Burgi Members (Figs. 3–4, Supplementary Fig. 4) are characterized by variable lithofacies changes, both laterally and vertically in section. In addition, there are consistent lithological features that allow for the lateral correlation and stratigraphic association of the sedimentary sequences.

The lower aggregate in the study area is upper Burgi Member and is characterized by alternating beds of mudstones and sandstones. The mudstones contain abundant pedogenic

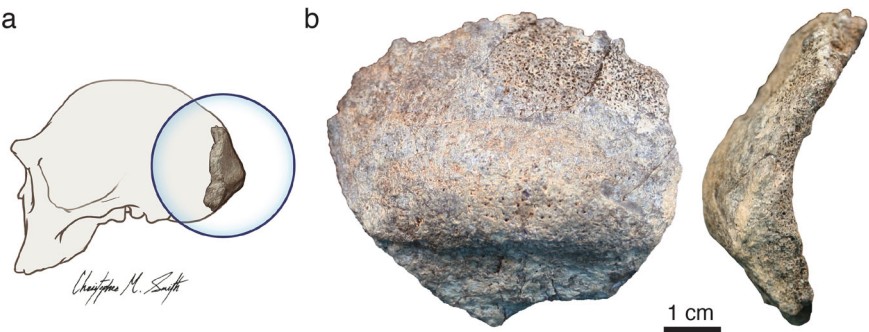

**Fig. 1 KNM-ER 2598 partial occipital. a** Inset image indicates the approximate anatomical location of the KNM-ER 2598 occipital. **b** Posterior view and right lateral view are shown.

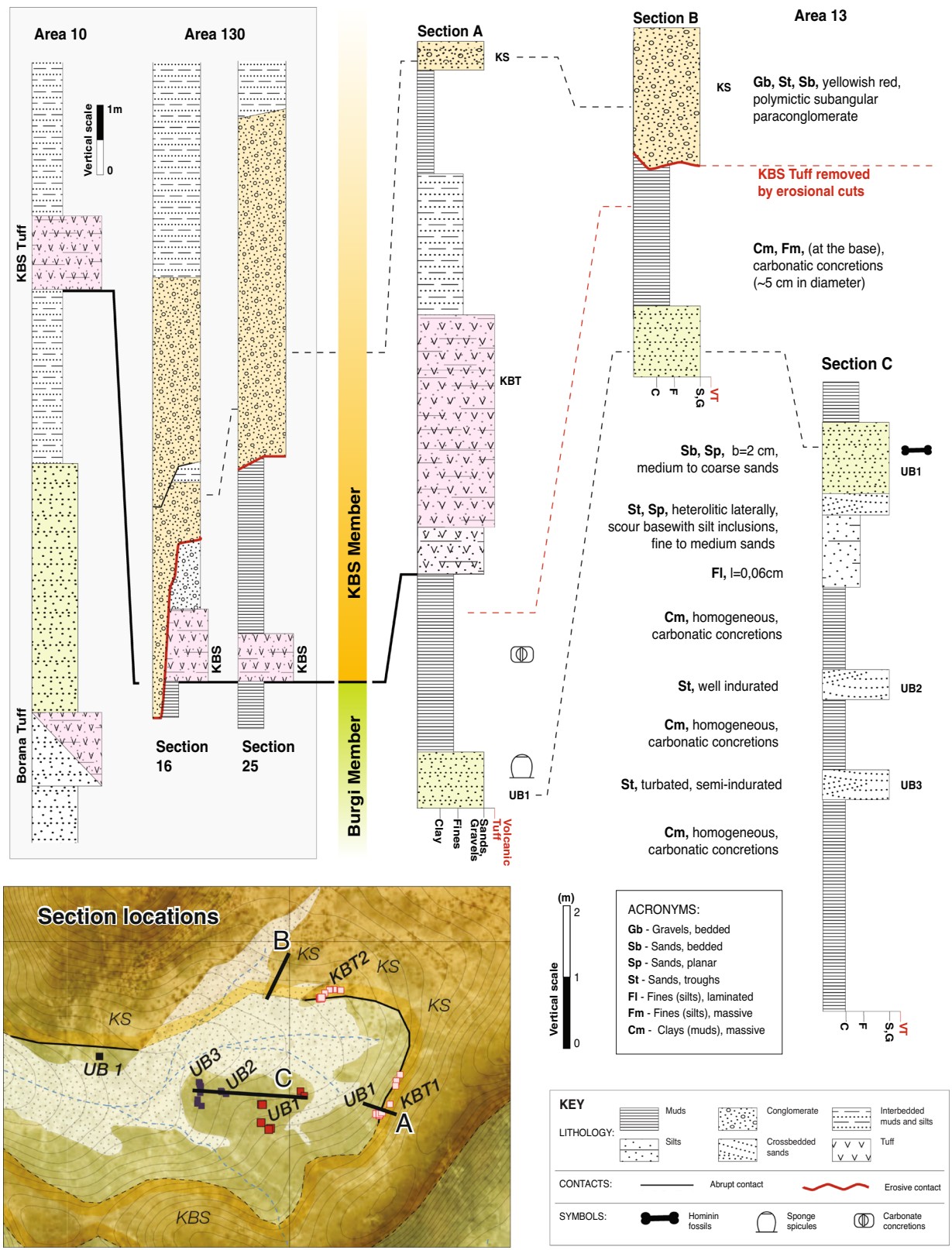

**Fig. 2 Stratigraphic sections within the study area and correlations with sections from nearby Area 10 and 130.** Note that the KBS Tuff (KBT) occurs in a gravel-silty unit (Section A) occasionally removed by erosion (Section B). In the absence of the tuff, this disconformity marks the boundary between KBS and upper Burgi deposits (Section B). Section locations are provided in the inset medallion map. See Fig. 3 for scale and orientation of inset map, and explanation of data points. Area 10 section is PNG-10.1 from ref. [17]. Area 130 sections are sections 16 and 25 from ref. [67].

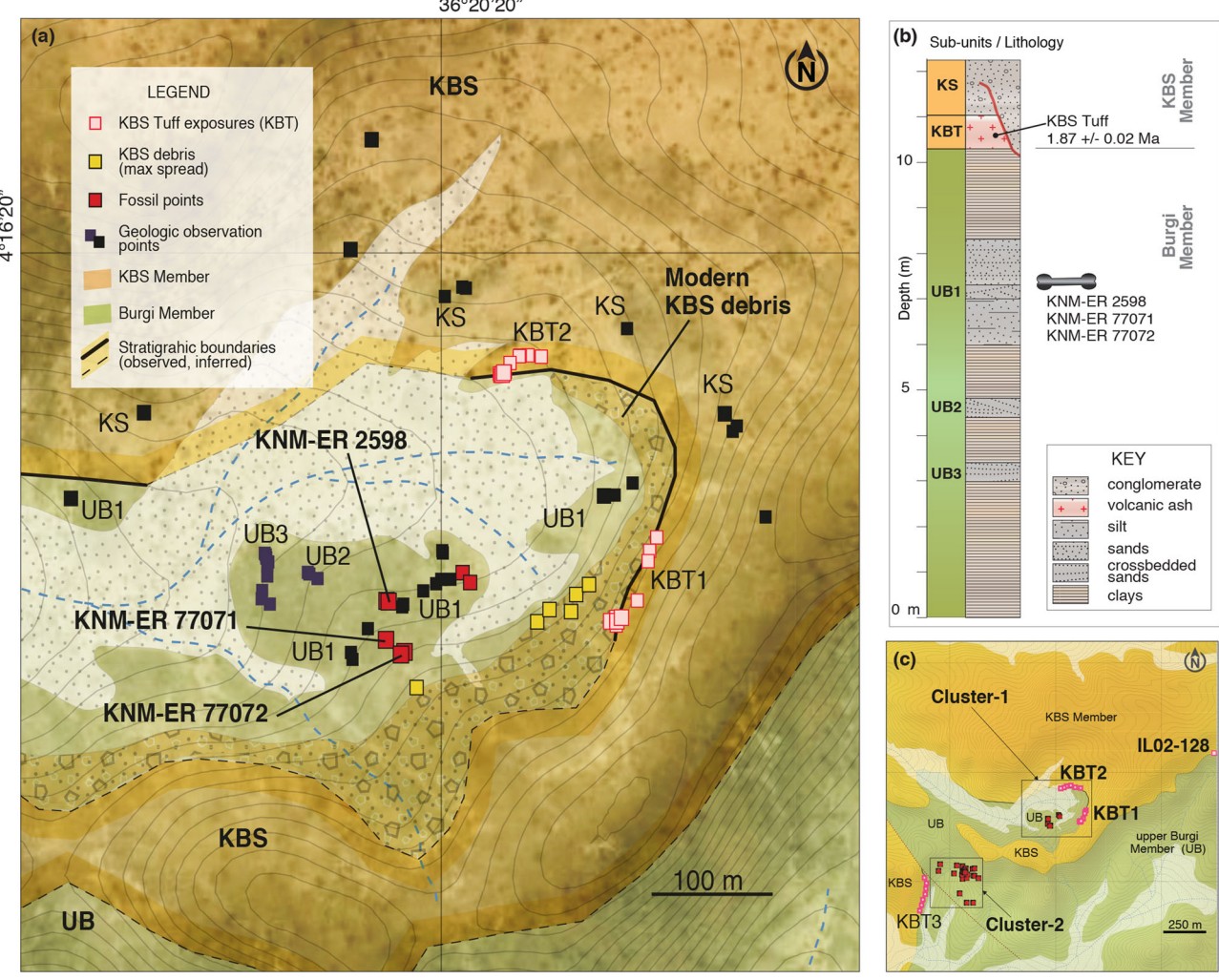

**Fig. 3 The KNM-ER 2598 fossil locality. a** Detailed map of the promontory that corresponds to upper Burgi Member fossil Cluster-1 in Area 13 (including KNM-ER 2598). Here we identify tuff outcrops using the field ID "KBTX" where X refers to the sequence of outcrops identified. Local exposures of the KBS Tuff (KBT1-KBT3) and the local lithological markers (sandstones KS, UB1, UB2, and UB3) are indicated. The contour interval is 0.5 m between topographic lines. **b** Litho-stratigraphic column for the sedimentary exposures in the study area. **c** General view of locations where the upper Burgi fossils were collected. The nearby KBS Tuff location sampled by Gathogo and Brown[15] is indicated (IL02-128). Map was created using ESRI ArcGIS Pro (version 2.1). The source data underlying Fig. 3 are provided as Supplementary Data 1.

carbonate nodules and fine silt laminae. There are three sandstone beds in this lower aggregate. The lower two beds (UB2 and UB3 in Figs. 2–3) are laterally restricted and characterized by trough cross-bedding indicative of fluvial deposition. The uppermost sandstone (UB1) is underlain by a laminated siltstone which coarsens upward into a fossiliferous cross-bedded sandstone. The uppermost beds attributed to the Burgi Member consist of an upward-fining sequence of silts and muds. The sandstones in this aggregate (UB1-3) are heterolithic but are petrographically similar (Fig. 4). The most common grains are poly- and monocrystalline quartz and feldspars (microcline and plagioclase), with smaller amounts of mica and other silicates. Bioclasts are present in the form of diverse siliceous spicules from freshwater megascleres sponges (Fig. 4). New hominin fragments found at the KNM-ER 2598 site (see below) were associated with the UB1 sandstone in the lower aggregate.

The upper aggregate in the study area is the KBS Member. The KBS Member sandstone (hereafter KS) is compositionally distinct, coarser, and thicker than those found in the upper Burgi Member (Fig. 4). The KS sandstone has an erosive base

locally grading into a subangular yellow-red matrix-supported polymictic conglomerate (Fig. 2). KS can be differentiated from the underlying UB1-3 both macro- (Supplementary Fig. 4) and microscopically (Fig. 4). Detailed information on the UB1, UB2, and KS minerology and sandstone microstructure are provided in Supplementary Note 1.

Survey revealed two locations in the upper Burgi Member sediments with fossils exposed on the surface (e.g., Cluster-1 and Cluster-2; Fig. 3). Cluster-1 (which surrounds the reconstructed KNM-ER 2598 discovery site) is located at the end of a promontory, in a larger depression where a modern rain drainage system eroded the KBS and younger units. Volcanic ash outcrops attributed to the KBS Tuff, and sandstone beds corresponding to upper Burgi and KBS Members, can be traced for several hundred meters on the rim of the depression. Cluster-1 was measured as approximately 3 m below the KBS Tuff, which is consistent with previous interpretations placing this surface at ~4 m below the tuff[1,7]. Site deflation was excluded as a possibility by examination of the sandstones and rock debris within 50 m of the cairn. The rocks present on the surface are exclusively associated with those of the upper Burgi Member based on mineralogy (Fig. 4) and

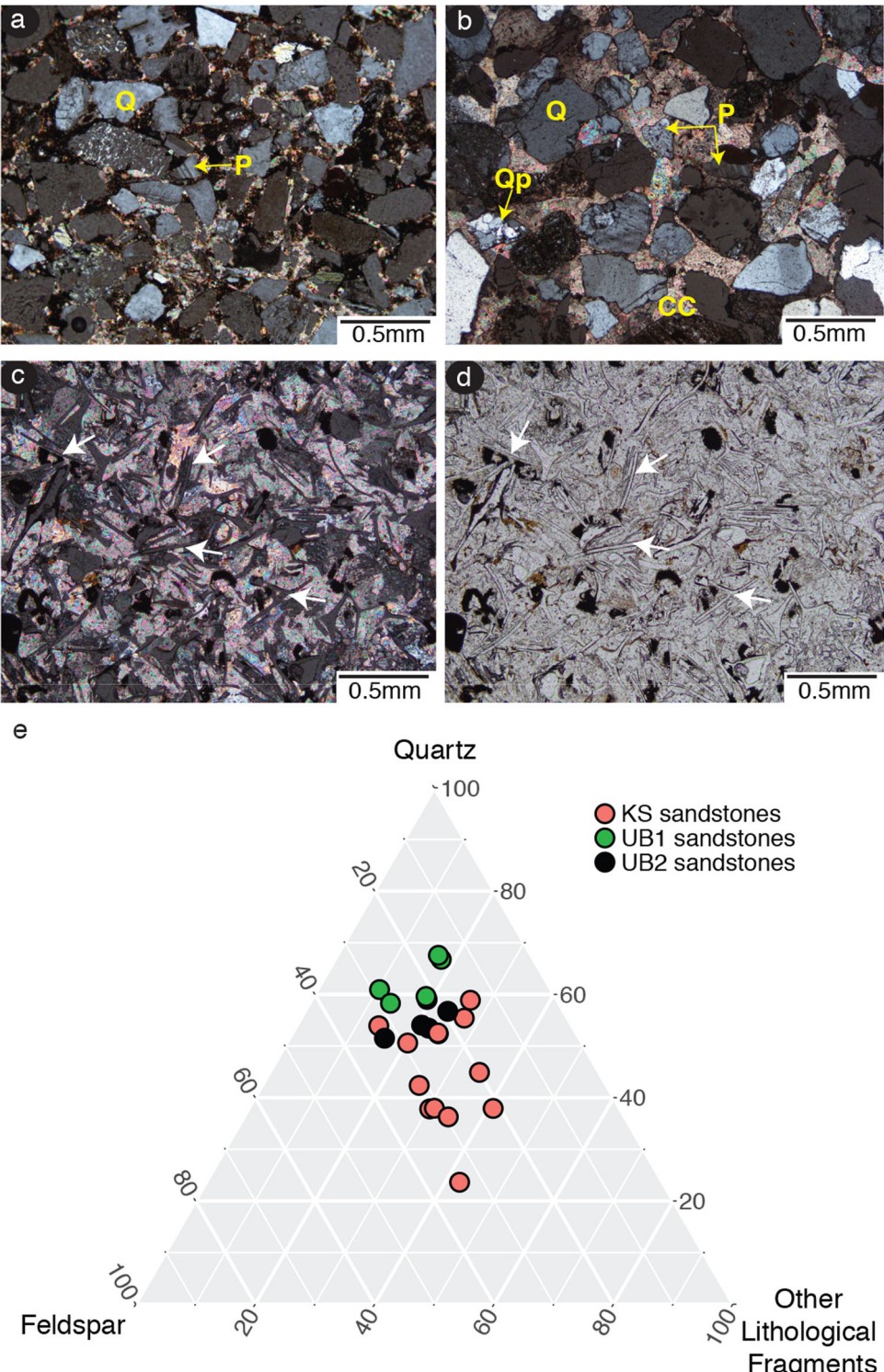

**Fig. 4 Mineralogy of Burgi Member ($n = 11$) and KBS Member ($n = 12$) sandstone thin-sections from Area 13. a** The UB1 sandstone lamination fabric as seen through the parallel arrangement of minerals with subrounded moderately sorted grains embedded in a calcite cement in cross-polarized light. **b** KS sandstone occurrences showing poorly sorted, angular grains of an immature sediment with distinct mineralogy as demonstrated by the presence of igneous rock fragments. UB2 sandstone spicules of megascleres sponges with monaxon and triaxon forms in **c** cross-polarized and **d** plane-polarized light. **e** A QFL ternary plot showing percent composition of upper Burgi (UB1, UB2) and KBS (KS) sandstones sampled in Area 13. CC: calcite cement, P: plagioclase, Q: quartz, Qp: polycrystalline quartz, white arrows: examples of sponge spicules. The source data underlying Fig. 4e are provided in the Source data file.

stratigraphic context (Figs. 2–4). That is, the overlying KS sandstones (i.e., polymictic subangular sandstones with granules larger than 2 cm), or fragments of KS, are not present at the Cluster-1 location and there is no remnant nor evidence of the sand and clay/silt units that sit between the UB1 sandstone and the KBS Tuff. Cluster-2 is located on an upper Burgi Member deposit adjacent to a third volcanic ash outcrop (KBT3 in Fig. 3).

The age range of the KBS Tuff (1.876 ± 0.021 Ma[16]), incorporating the measurement error, is 1.855–1.897 Ma. The fossils from the UB1 sandstone originated below the KBS Tuff and must therefore be older than 1.855 Ma. This estimate should be interpreted as a conservative theoretical upper constraint of the age range given that the UB1 sandstones sit 4 m below the KBS Tuff. In regard to the lower constraint on the age range, the UB1 sandstones and overlying fine-grained sediments of the upper Burgi Member can be correlated to a published lithologically-similar section in neighboring Area 10[17]. This correlation would assign the UB1 unit and the fossils deriving from it to a level above the Borana Tuff. However, the age of the Borana Tuff is as yet unknown. Correlations to other lateral tuff markers in the Shungura Formation are not completely resolved (see the "Discussion" section), allowing only a qualitative assignment in the proximity of the base of the Olduvai Subchron (1.934 Ma), but not excluding a slightly older age.

**Fossils**. Hominin cranial vault fragments, a partial ilium, and a proximal 3rd metatarsal (MT3) were collected from the upper Burgi Member of Area 13. Whereas the vertebrate fauna was collected in two fossiliferous clusters (the previously mentioned Cluster-1 and -2; Fig. 3), all hominin material was recovered within Cluster-1 (i.e., <50 m of the KNM-ER 2598 cairn; see also Supplementary Data 1). All hominin fossils were weathered surface finds that appear to have been sitting exposed on the surface for several years. A direct association with KNM-ER 2598 could not be established. As such, all of the hominin specimens recovered here were issued distinct NMK accession numbers.

Five small fragments that are likely to have originated from a hominin cranial vault were recovered (Supplementary Fig. 5; KNM-ER 77066, KNM-ER 77067, KNM-ER 77068, KNM-ER 77069, KNM-ER 77070). All 5 of the fragments preserve unclosed suture borders as in KNM-ER 2598 and display the fairly divergent sutural limbs characteristic of *Homo erectus* crania[3]. However, none of the cranial fragments directly refit with KNM-ER 2598 nor can they be linked definitively vis-à-vis surface texture and coloration. More information on these non-diagnostic fragments is provided in Supplementary Note 2.

KNM-ER 77071 is an abraded hominin left proximal MT3 (Fig. 5). This specimen was located 29 m from the KNM-ER 2598 cairn. This fragment preserves the proximal shaft and the base, including discernable contact facets for the MT4 laterally and the MT2 medially. The MT4 contact facet is the larger contact facet and is bounded by a deep gulley inferiorly. There is a partial plantar process visible on the plantar aspect of the base. The base is 17.8 mm superoinferiorly in its maximum dimension, with a 10.9 mm superior border and a 5.6 mm inferior border. The metatarsal shaft measures 9.1 mm superoinferiorly by 6.8 mm mediolaterally at the cross-section break. KNM-ER 77071 is hominin-like in having a dorsoplantarly tall base relative to the width[18], flat base, and intermetatarsal contact facets[19], and in lacking the medial and lateral indentations for the transmission of intermetatarsal ligaments that are characteristic of apes[20]. The complete MT3 would have been slightly larger and more robust than the OH8 proximal MT3. Like other hominin MT3s, such as *H. erectus* specimens D2021[21] and KNM-ER 803[22], KNM-ER 77071 has a single weak dorsal contact for articulation with MT2.

There are no characters preserved on this proximal MT3 that morphologically or functionally distinguish it from other Plio-Pleistocene hominins (Fig. 5).

KNM-ER 77072 is a hominin partial ilium (Fig. 6). This specimen was located 40 m from the KNM-ER 2598 cairn. The specimen preserves most of the iliac tuberosity, greater sciatic notch region, and much of the body, although the ala is broken off anteriorly and superiorly. The ilium, as preserved, measures 84.0 mm anteroposteriorly and 55.5 mm superoinferiorly. There is no iliac pillar visible, although this may have been present and more anteriorly situated (Fig. 6). The pear-shaped sacroiliac joint in KNM-ER 77072 is well preserved and posteriorly bounded by a deep postauricular groove and a mound of bone on the most dorsal region of the iliac tuberosity. The iliac tuberosity is 20.6-mm thick. The sacroiliac joint measures 43.0 mm super-oinferiorly and ~26 mm across the widest portion. The greater sciatic notch is wide and, although incomplete distally, would likely have had a sciatic notch angle >100°. The sciatic notch, as preserved, is 28.7 mm across and ~14-mm deep, and has a shallow appearance. The gluteal surface is well preserved and a line demarcating the boundary between *gluteus medius* and *g. minimus* origins can be observed.

Qualitative comparisons align the morphology of KNM-ER 77072 with genus *Homo*. The specimen differs from australopiths in robusticity of the ilium, especially the iliac tuberosity[23] and acetabulosacral pillar[24]. KNM-ER 77072 is most easily compared with ilia associated with *Homo erectus* (i.e., KNM-ER 1808 and KNM-WT 15000) or attributed to the taxon (i.e., UA 173/405, BSN49/P27, OH 28, KNM-ER 3228). Most potential *Homo erectus* ilia (excluding OH 28 and KNM-ER 3228) share the following features with KNM-ER 77072: dorsally thick iliac tuberosities, weakly developed gluteal muscle markings, a moderately thick acetabulosacral buttress, shallow and wide sciatic notches, auricular surfaces that appear fairly small, and a relatively anterior origin of the weakly-developed iliac pillar (if preserved). KNM-ER 77072 is similar to UA 173 in possessing a deep postauricular groove and thick dorsal portion of the iliac tuberosity[23]. There are no pelves formally associated to *Homo habilis* or *Homo rudolfensis*. However, the diminutive KNM-ER 5881 pelvis (Fig. 6, Supplementary Fig. 7) has been suggested to belong to a "non-*erectus*" species of *Homo*[25]. Although incomplete, KNM-ER 5881 preserves an iliac pillar that originates quite posteriorly but is directed anteriorly[25], reflecting a different pelvic geometry from KNM-ER 77072.

Thirty-two non-hominin taxonomically identifiable fossils were collected from the upper Burgi Member of Area 13 (Fig. 7a, b), including bovids ($n = 17$), equids ($n = 4$), suids ($n = 3$), cercopithecoids ($n = 3$), a proboscidean ($n = 1$), a hippopotamid ($n = 1$), a rhinocerotid ($n = 1$), a giraffid ($n = 1$), and a snake (i.e., *Serpentes*; $n = 1$). The fossil identifications suggest that the fauna represents primarily $C_4$ grazers, which was confirmed by isotopic analysis from 17 enamel samples. Carbon isotope data for the enamel samples range from −1.8 to +2.5‰ (median = +1.4‰; Table S2, Fig. 7c). Oxygen isotope values (Fig. 7d) range from −3.5 to +3.6‰ (median = +0.1‰). The single hippopotamid had the most depleted value and an alcelaphin bovid had the most enriched value.

## Discussion

Our fieldwork found a discrepancy between the historical records of collecting areas in East Turkana and the modern, formalized boundaries of these areas. The East Turkana collection areas are principally defined by landscape features (e.g., ephemeral streams) and have been well-mapped (e.g., Supplementary Fig. 1) since the mid-1980s[26]. The KNM-ER 2598 coordinates,

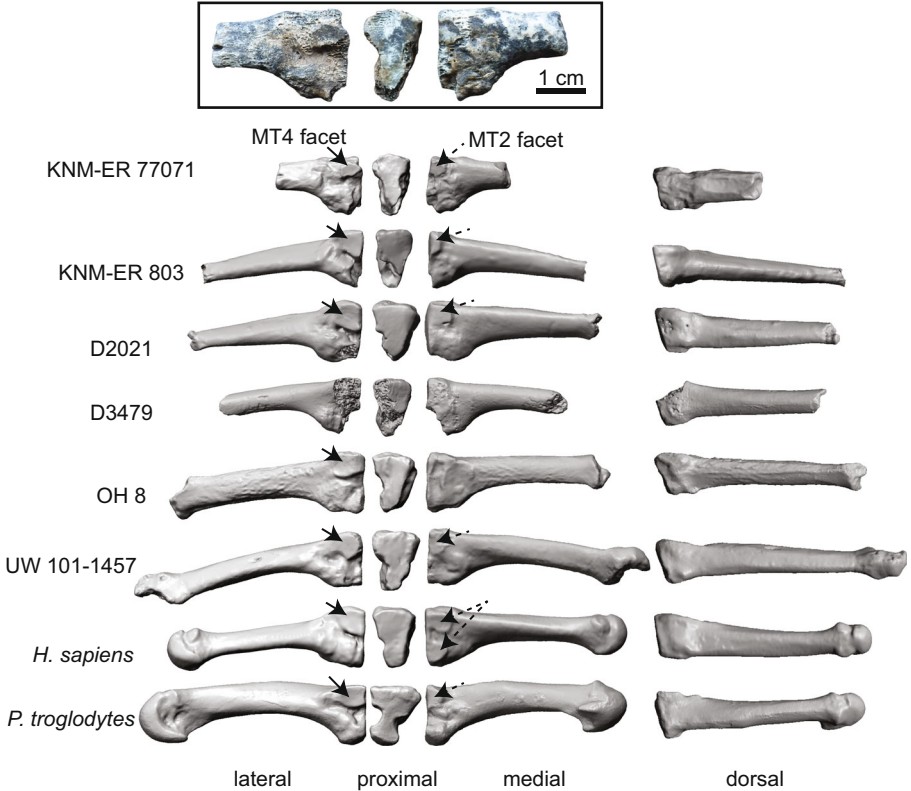

**Fig. 5 Proximal 3rd metatarsal anatomy.** Proximal left the third metatarsal KNM-ER 77071 is inset to show lateral, proximal, and medial views. Comparative MT3s for *Homo erectus* (KNM-ER 803, D2021, D3479), *Homo habilis* (OH 8), *Homo naledi* (UW 101-1457[68]), humans, and chimpanzees are shown below in lateral, proximal, medial, and dorsal views. MT4 facets indicated by solid arrows, and MT2 facets indicated by the dashed arrows. Like the other hominin fossils, KNM-ER 77071 has only a single dorsal MT2 facet on the medial side. Metatarsal models scaled to the same height of the base for visual comparison. UW 101-1457, D2021, and D3479 images are mirrored for consistency.

determined through aligning 1970s aerial photos with modern satellite imagery, are located in what is defined as Area 13. This finding contradicts the original publication[6], which listed Area 15 as the location where KNM-ER 2598 was collected. Historical records are lacking but photographs from the 1974–1975 field seasons confirmed that collections were taking place in the location that is currently designated as Area 13 (Supplementary Fig. 2). Area 15 is now more widely recognized as being principally composed of Lonyumun Member sediments (~4.0–4.3 Ma)[15,27], and is therefore unlikely to preserve Early Pleistocene sediments and fossils. In contrast, Area 13 has recently produced a number of other early *Homo* specimens attributed to the upper Burgi and KBS Members[17,28,29]. One of these, a *Homo habilis* dentition (KNM-ER 64060), originated <1.5 km from this location and is dated to ~2.0 Ma[17], documenting the close temporal and geographic proximity of early *H. erectus* and *H. habilis* in East Turkana.

The discrepancy in East Turkana collections recorded as Area 15 versus Area 13 almost certainly extends to other vertebrate fossils recovered in the 1970s. The publicly-available Turkana Basin Database[30] records 44 distinct specimen numbers for Pleistocene fossils from collection Area 15. These fossils are likely to derive from other collection areas based on our current understanding of the collection area boundaries and geology contained within these regions. To our knowledge, the only documented vertebrate fossils which originate in Area 13 are those reported in this study. Further investigations that combine archival information and modern reconnaissance are needed to establish the provenience of the 1970s fauna reported from Area 15.

The most straightforward interpretation of the geological data presented here is that the new hominin fossils, and presumably KNM-ER 2598, weathered out of the UB1 sandstone and lay exposed on the surface until they were discovered. We found no evidence of deflation at the KNM-ER 2598 cairn location and surrounding Cluster-1 area. The sandstones and sandstone fragments found in Cluster-1 have distinctive sedimentary features (i.e., trough and planar cross-bedding) aligning them with UB1 and, furthermore, are petrographically distinctive from the younger sandstones (KS) overlying the KBS Tuff. A mixture of surface sandstones was found about 200 m away from the KNM-ER 2598 cairn location, in locations where torrential runoff moves the KS sandstones into drainage areas overlying Burgi Member outcrops (e.g., "modern KBS debris" shown in Fig. 3). Although we cannot completely exclude the possibility that these hominin fossils are derived from younger sediments, the hypothesis that the fossils on the Cluster-1 surface could result from deflation of a younger sedimentary package that has subsequently eroded away is not supported by any of our observations.

The KBS Tuff (1.876 ± 0.021 Ma[16]) acts as an upper (=minimum) age constraint to KNM-ER 2598 and the fossils described here. The maximum age of these fossils is currently unresolved. Lacking lower (=maximum) age constraints in our study area, we correlated the UB1 sandstone and overlying fine-grained sediments of the upper Burgi Member with a nearby lithologically-similar sedimentary succession[17], which tentatively aligns the UB1 unit (and associated fossils) to a level above the Borana Tuff. This discontinuous tuff occurs within a complex stratigraphic succession and past correlations resulted in conflicting, often contradictory, stratigraphic positions[31,32] and association with

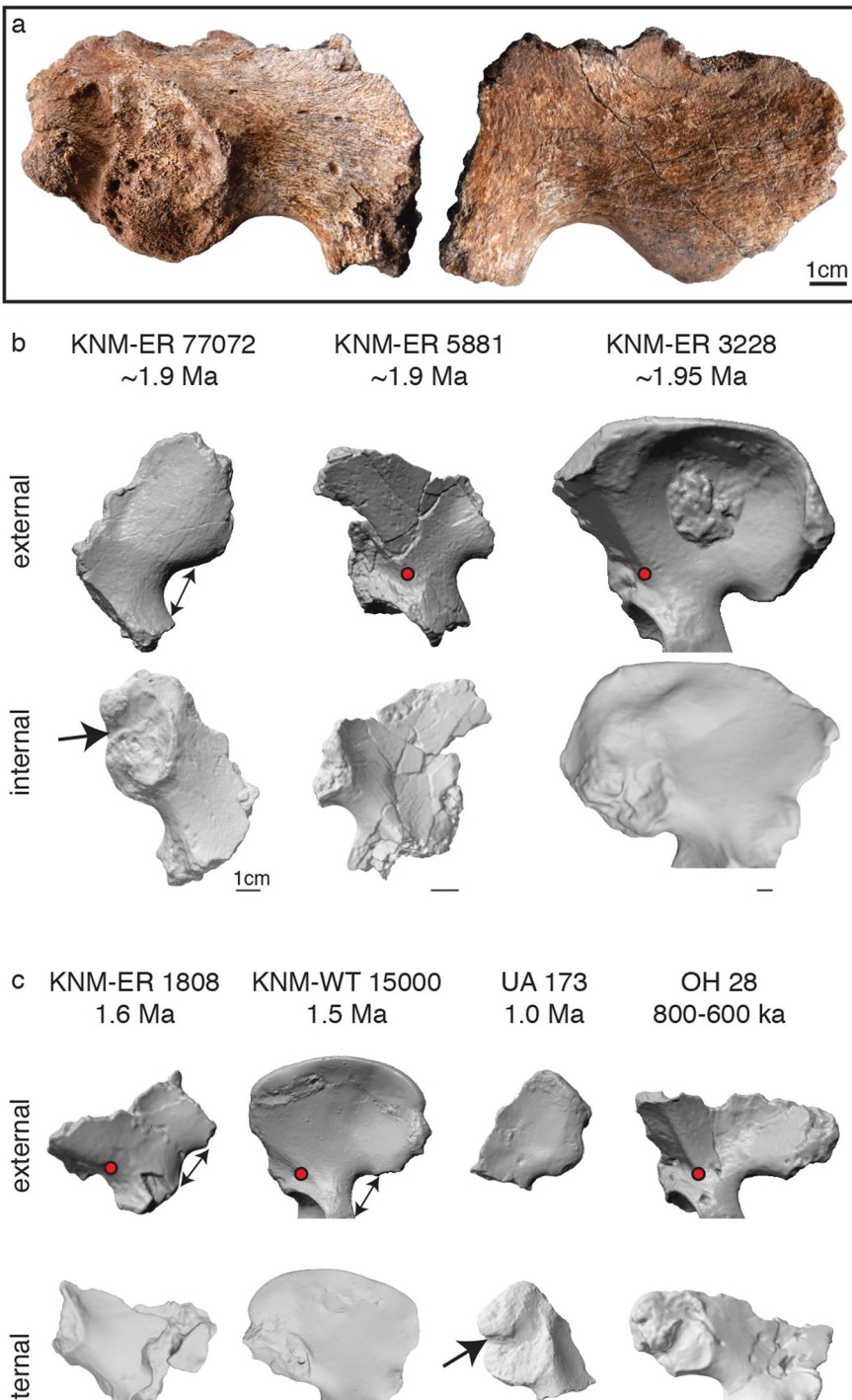

**Fig. 6 Ilium anatomy. a** Left partial ilium KNM-ER 77072 shown in medial and lateral views. **b** Scanned models of contemporaneous ilia from East Turkana show differences in absolute size and iliac pillar configuration. Note that diminutive KNM-ER 5881 has an iliac pillar (red dot indicating iliac pillar base) that is modest in thickness but originates posteriorly relative to the anterior border of the ilium, whereas KNM-ER 3228 has a massive iliac pillar that originates more anteriorly. Upper Burgi specimen detail is also shown in Supplementary Fig. 7. **c** *Homo erectus* ilia available for comparative study. KNM-ER 77072, KNM-ER 1808, KNM-WT 15000, and UA 173/405 share the following features, when preserved: thick dorsal regions of the iliac tuberosity, weak muscle markings on the gluteal surfaces, a moderately thick acetabulosacral buttress, wide and shallow greater sciatic notches (indicated by double-ended arrow), auricular surfaces that are small compared to *Homo sapiens*, deep postauricular grooves (bold arrow), and a relatively anteriorly-positioned and weakly-developed iliac pillar. KNM-ER 3228, KNM-WT 15000, UA 173, and OH 28 images are mirrored for consistency. KNM-WT 15000 was scanned from a cast that included a reconstructed iliac crest. 1-cm scale is shown below each fossil.

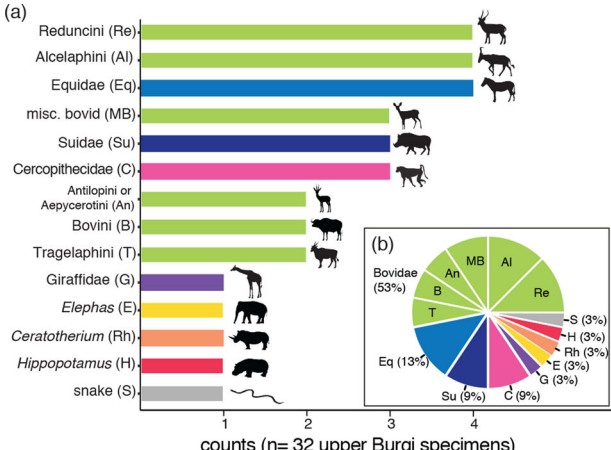

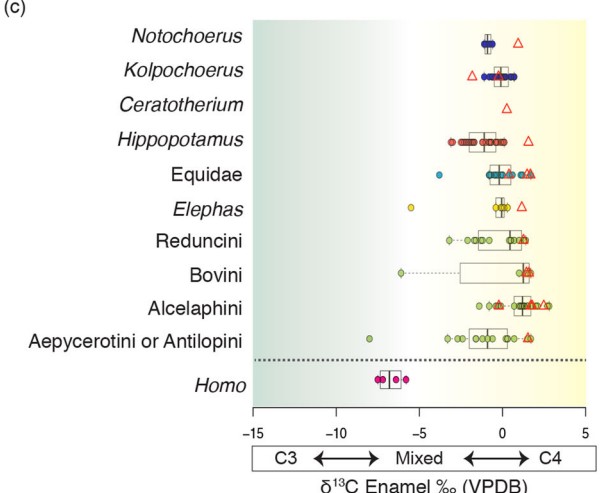

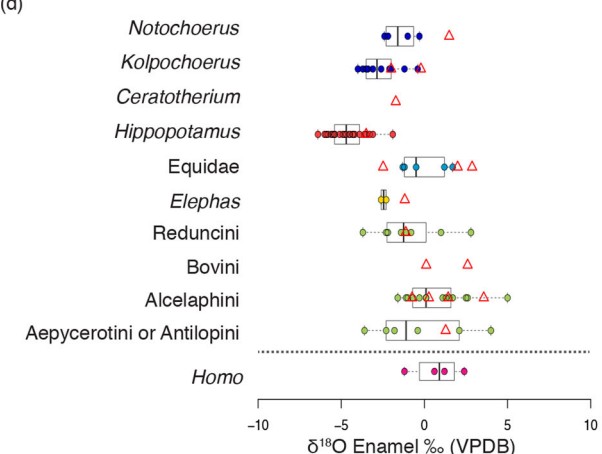

**Fig. 7 Non-hominin fossils recovered in Area 13 from 2017 to 2019.**
Thirty-two taxonomically identifiable vertebrate fossils were collected (**a**, **b**). Miscellaneous bovids are those specimens for which a specific tribe could not be determined. Antilopini and Aepycerotini cannot consistently be differentiated based on isolated molars and are pooled together. **c**, **d** New and existing upper Burgi Member enamel isotopic data. New enamel isotopic data from Area 13 (red triangles) are not incorporated into boxplots. Median values are represented by vertical lines within the box, the edges of the boxes represent quartile ranges, horizontal dashed lines represent the range and outlier values are plotted as circles outside of the range. Raw δ¹³C and δ¹⁸O, presented as circles superimposed on the box plots, are consistent with open habitats that had $C_4$-dominated diets and locally arid conditions, and with the upper Burgi isotopic signatures more broadly. Silhouettes are from http://phylopic.org, with attribution to A. Venter, Herbert H. T. Prins, David A. Balfour, and Rob Slotow (vectorized by T. Michael Keesey) for Reducini, Alcelaphini, Tragelaphini, and miscellaneous bovid (https://creativecommons.org/licenses/by/3.0/). The source data underlying Fig. 7 are provided in the Source data file.

base of the Olduvai normal subchron (1.934 Ma[9,10]). The maximum age constraint could eventually be resolved if the magnetic polarity of the UB sandstones and of the Borana Tuff are clarified.

The newly recovered hominin metatarsal and ilium are consistent with genus *Homo*. The flat MT3 base and flat contact facet for articulation with the MT4 suggest limited mobility at both the tarsometatarsal joint and lateral intermetatarsal joint, indicating that a transverse arch is present in the foot of KNM-ER 77071. However, KNM-ER 77071 lacks an MT3 plantar facet for contact with the MT2, as is also the case for other *Homo* specimens (i.e., OH 8, D2021, KNM-ER 803). The lack of a plantar facet on the MT3 has been proposed to have allowed moderate cuneo-metatarsal mobility[21]. It may also reflect a slightly lower apex of the transverse arch than in *Homo sapiens*, whereby the MT2 and MT3 would only make contact along the dorsal portion of their bases. This may suggest a slightly more mobile transverse arch in earlier *Homo* than in *Homo sapiens*, but testing this idea is contingent on the discovery of more complete hominin feet.

The KNM-ER 77072 partial ilium adds to the growing list of pelvic specimens that are likely attributed to *Homo erectus* (Fig. 6). KNM-ER 77072 is similar in size and morphology to KNM-ER 1808, KNM-WT 15000, and UA 173—specimens which most workers agree are *Homo erectus*. KNM-ER 77072 also appears generally consistent with the morphology described for the probable *Homo erectus* pelvis from Gona (BSN49/P27)[36]. These pelves share, among other features, dorsally thick iliac tuberosities, weak muscle markings on the gluteal surfaces, shallow and wide sciatic notches, and fairly small auricular surfaces. Wide and shallow greater sciatic notches are seen in earlier taxa like *Australopithecus*, so this cannot be excluded as a plesiomorphic character (as opposed to indicating female sex or increased obstetric demands; see also ref. [36]). These fossils collectively point to an ilium that is only moderately robust in *Homo erectus*, posing a challenge for including the absolutely large and robust specimens from eastern Africa (i.e., OH 28, KNM-ER 3228) within the *Homo erectus* hypodigm. The link between these fossils and *Homo erectus* has always been tenuous; OH 28 is assumed to be *Homo erectus* based on association with a *Homo erectus*-like femoral diaphysis[37], and KNM-ER 3228 has been aligned with *Homo erectus*[38,39] based on similarities with OH 28 (but has also been considered a candidate for *Homo rudolfensis*[40]; see below). Additional pelves associated with *Homo erectus* craniodental material, ideally sampling multiple individuals from a single locality such as Dmanisi, would clarify the range of morphological variation we can accept in this taxon.

both normal and reverse paleomagnetic polarity intervals[17,33–35]. A conservative approach[31] correlated the Borana Tuff with "an unnamed tuff in the upper G Member" of the Shungura Formation. This assignment does not exclude a reverse polarity (C2r.1r, beginning of the Olduvai Subchron) for the tuff[35] that would expand the range of the UB1. If these East Turkana fossils were deposited in the C2r.1r chron, they would be contemporaneous with the oldest *Homo erectus* specimen from Drimolen (DNH 134). We consider this scenario unlikely given that DNH 134 is constrained within the reversed subchron (1.934–2.120 Ma) and these East Turkana fossils lay just a few meters below the KBS tuff (1.876 ± 0.021 Ma[16]), likely toward the

KNM-ER 77072 also confirms that diversity in pelvic morphology is present in genus *Homo* at ~2 Ma, hinting at postcranial differences that may accompany the taxonomic diversity present in East Turkana. There are at least three species of *Homo* in East Turkana during the brief interval from 1.85 to 1.95 Ma: *Homo habilis*, *Homo rudolfensis*, and *Homo erectus*[41]. There are also now three morphologically distinct pelvic specimens from this same interval: KNM-ER 5881[25], KNM-ER 3228[38,42], and KNM-ER 77072 (Fig. 6, Supplementary Fig. 7). The KNM-ER 5881 ilium (~1.9 Ma) differs morphologically from other fossils and is associated with a femur that is similar to *Homo habilis* (i.e., OH 62) in cross-section[25]. The femur cross-sectional shape, along with the pelvic morphology, has led to the conclusion that KNM-ER 5881 is likely attributed to either *Homo habilis* or *Homo rudolfensis* (for which postcranial morphology is entirely unknown)[25]. Notably, KNM-ER 5881 has a posteriorly originating iliac pillar, whereas there is no evidence of the iliac pillar in what is preserved of KNM-ER 77072. This could mean that a discernable iliac pillar was not present in KNM-ER 77072 or, more likely, that it was weakly developed and more anteriorly positioned in this specimen. Whereas KNM-ER 5881 is diminutive, KNM-ER 3228 (~1.95 Ma) is from a large and heavily-muscled individual with a marked gluteal surface concavity. It would require *Gorilla*-like or *Pongo*-like levels of body size dimorphism for KNM-ER 5881 and KNM-ER 3228 to represent male and female individuals of the same species[25]. Moreover, it appears that all three of these contemporaneous fossils sample individuals of different body size based on the lower ilium length and the iliac tuberosity size (Supplementary Fig. 7). Finally, although greater sciatic notch angle is sexually dimorphic in modern humans (but not necessarily pre-human hominins[43]), KNM-ER 3228 apparently represents the first example of a narrow (male or 'masculine') sciatic notch in the fossil record, a condition not clearly detected again until late *Homo*. If KNM-ER 77072 is *Homo erectus*, the disparities in size and morphology among these contemporaneous specimens would lend support to the idea that KNM-ER 5881 and KNM-ER 3228 derive from species other than *Homo erectus*. The alternative possibility is that some (or all) of these specimens belong to a single, postcranially variable species of *Homo*. The pelvic diversity seen in the Turkana Basin during this narrow time interval suggests there was substantial selective pressure operating on pelvic and hip function fairly early in the evolution of *Homo*.

It is possible that the hominin specimens described here come from a single individual given the spatial proximity in which they were all recovered. None of the newly recovered hominin cranial fragments directly refit with KNM-ER 2598, although they are morphologically consistent with KNM-ER 2598 and other *Homo erectus* specimens (Supplementary Figs. 5–6). The possibility that the newly recovered hominin fragments reported here are associated with KNM-ER 2598 is even more plausible when considering that our large surface survey of the upper Burgi Member deposits produced only a few dozen taxonomically identifiable fossils.

Multiple lines of evidence suggest that this locality was near a well-watered and open, grassy environment. The non-hominin fossil taxa recovered in Area 13 are primarily hypsodont $C_4$ grazers with essentially no mixed feeders. Carbon isotope values fall within the range, or in some cases slightly outside the range, of those reported for the same taxa from the upper Burgi Member (Fig. 7c; refs. 44–47). The oxygen isotope values are generally more positive than other upper Burgi Member samples, including a single hippopotamid, which potentially indicates more evaporated local source waters or more arid conditions than other upper Burgi Member locations (Fig. 7c). The presence of sponges in the UB1 sandstone indicates a long-term, stable body of water

at this locality[48]. The non-hominin fossil taxa and the limited suite of isotopic data represent a fauna associated with open habitats that had $C_4$-dominated diets. Existing paleosol carbonate data indicate significant variation in woody cover during the upper Burgi Member (between ~10 and 65%)[45,49–51]. It is possible that the new enamel data reflect a more open subset of this heterogeneous environment.

In summary, new investigations at the KNM-ER 2598 locality have produced several key findings. The newly discovered hominin postcranial elements include a partial ilium and a proximal metatarsal. Although neither element can be definitively assigned to the same individual as KNM-ER 2598, the ilium and metatarsal are morphologically consistent with *Homo erectus*. The KNM-ER 2598 locality is located in a different East Turkana collection area than initially reported in the 1970s, which may have resulted in incorrect interpretations of both the hominin and faunal material over the last few decades. The new fauna consists primarily of $C_4$ grazers and suggests a fairly open paleoenvironment. This study confirms that the location where KNM-ER 2598 was discovered is associated with distinct sandstones that are exclusively associated with the upper Burgi Member. The KNM-ER 2598 site is stratigraphically positioned ~3 m below the KBS Tuff, requiring that the fossils from this location are >1.855 Ma. The KNM-ER 2598 occipital, as well as the new ilium and metatarsal reported here, are among the oldest fossil specimens likely attributable to *Homo erectus*.

## Methods

**Identification of the locality**. East Turkana fossil locations in the 1970s were originally recorded on aerial photographs that continue to serve as records of where fossils were collected (Supplementary Fig. 1). The aerial photographs that documented the KNM-ER 2598 find were captured in 1970 by Hunting Surveys, Ltd. and were housed at the National Museums of Kenya at the time of study. Fossil finds were marked on the photographs with pinpricks and with corresponding field numbers recorded on the back of the aerial photo. We used Google Earth imagery to approximate the geospatial location of KNM-ER 2598 in geographic coordinates (Supplementary Fig. 1). Photographs from the 1974–1975 field seasons when KNM-ER 2598 was recovered were provided by Tim White (University of California-Berkeley). These photos support our assertion that this locality, which was relocated from aerial imagery, corresponds to the 1974–1975 field campaign locations (Supplementary Fig. 2).

**Geological context**. First, the locality was surveyed for exposed outcrops and previously described sections were investigated[15]. Two main fossil clusters were found, one at the KNM-ER 2598 locality (Cluster-1; Fig. 3) and at a second location nearby (Cluster-2). Volcanic ash demarcates the boundary between the upper Burgi and KBS stratigraphic units and serves as a reference to identify lithological marker horizons in both units (Fig. 3). These lithological elements were then described both in outcrop and in thin sections and subsequently assigned to particular facies. Geological specimen description and logging of stratigraphic section were adapted from standard techniques[52,53]. Descriptions included grain size, color, bed shape, lateral variation in the bed, sedimentary structures, and bed-top and -bottom interactions, as well as any post-depositional features. Thicknesses were measured with Jacob's staff, which was also used to note prominent stratigraphic boundaries and document lithology across the study area. Structural measurements (strike, dip, and dip direction) were also recorded on bed surfaces to supplement mapping efforts in the study area and note any stratigraphic distortions. Outcrops of tuff and sandstone marker beds (both thickness and horizontal extent) were recorded for geological mapping following ref. 26. The geographic coordinates of the outcrops were acquired using GPS systems. Outcrops and boundaries were then plotted on a topographic map extracted from a digital surface model, acquired from Apollo Mapping World-DEM. We calculated the average bedding by interpolating data from three separate locations with precise geographic coordinates (i.e., 3-point problem). The Burgi-KBS boundary was extended beyond our observation points by intersecting the topography with the averaged bedding plane (221/08).

After regional lithological descriptions were completed, microstratigraphic work was conducted to better contextualize the KNM-ER 2598 locality. Microstratigraphy was aimed at defining the provenance of surface sandstones and placing these units within the overall geology of the area. We also noted post-depositional features associated with the locality which includes colluvial wash and deflation. In situ sandstones from the upper Burgi and KBS members were collected as reference material during stratigraphic descriptions to compare with the KNM-ER 2598 locality sandstones. Export of geological samples was conducted

through a material transfer agreement between the NMK, The George Washington University, and the University of the Witwatersrand, and authorized by the Kenyan Department of Mining. First, surface sandstones were described from petrographic thin section[54]. Twelve KBS Member and 11 Burgi Member samples were prepared for thin section at the University of the Witwatersrand School of Geosciences Rock Cutting Laboratory. The surface sandstones were matched with sandstones of known stratigraphic location to determine their provenance. This sourcing assisted in establishing the relative proximity and movement of these surface sandstones, enabling the study of post-depositional conditions such as overwash and deflation. Second, the mineralogical composition of the sandstones was characterized using QFL framework minerals (i.e., quartz, feldspar, other lithological fragments). Medium to coarse-grained channel sandstones from different parts of the area and at varying stratigraphic levels were sampled and point counted using the Gazzi–Dickinson method[55,56]. This involved selecting 350 random points in a single thin section and determining the mineralogy according to the QFL system. These mineralogical proportions were counted and converted into percentages before being plotted as a QFL ternary diagram to show mineralogical differences between Burgi and KBS sandstones (Fig. 4).

**Fossils.** Fossils were collected with the oversight and authority of the National Museums of Kenya (NMK) as mandated by Kenyan law. An Excavation and Exploration license (reference number NMK/GVT/2) was obtained from the Ministry of Sports, Culture, and Heritage through the NMK. The PI (Hammond) retained permits from the Kenyan National Commission for Science, Innovation, and Technology (NACOSTI; permits P/17/46866/17343 effective July 26, 2017; P/ 18/46866/25344 effective October 18, 2018) for field research.

We performed standard surface surveys in collection Area 13 over the course of three field seasons in 2017–2019. Surveys occurred in fossiliferous locations identified as Cluster-1 and Cluster-2 in Fig. 3. Additional intensive surface crawls were conducted within 50 m of the KNM-ER 2598 cairn. Vertebrate fossils were collected if they were identifiable as primate, mammalian cranial elements, horncores, mammalian teeth that were at least 50% complete, astragali, and long bones that were at least 50% complete and/or preserved at least one articular surface, and snake vertebrae. All fossils were found either sitting on the surface or partially embedded in sediments.

Sixteen of the fossil teeth were well-preserved enough to sample for stable carbon and oxygen isotopes. Seventeen samples total were collected, with two samples collected from an alcelaphine specimen represented by both upper and lower molars. Enamel sampling was performed at the National Museums of Kenya. Two to four milligrams of enamel powder was collected from cracks and breaks in each tooth using a high-speed drill, following protocols published elsewhere[44,57,58]. Export of the enamel powder for isotopic analysis was authorized by the NMK through a Material Transfer Agreement. Isotope data were analyzed as described in ref. [59] at the Stable Isotope Laboratory at Lamont-Doherty Earth Observatory. Data were quantitatively compared to published upper Burgi Member isotopic data[44–47,58,60–64] using non-parametric Kruskal–Wallis comparisons in R (Supplementary Table 2). We use −11.9 and +3.4‰ as $C_3$ and $C_4$ endmembers to calculate the percent $C_4$ diet from the tooth enamel data. These values are based on an atmospheric $\delta^{13}C$ value of −6.5‰ and biosynthetic fractionation factors for $C_3$ and $C_4$ plants[65]. $C_3$-dominated diets are those that include 0–25% $C_4$ vegetation (−12 to −8‰) mixed diets are >25% to <75% $C_4$ vegetation (>−8‰ to −0.5‰) and $C_4$-dominated diets are those with >75% $C_4$ vegetation (>−0.5‰).

The hominin metatarsal (KNM-ER 77071) and ilium (KNM-ER 77072) recovered in Cluster-1 were qualitatively compared with available hominin comparative material. Limited quantitative measures could be extracted due to the fragmentary nature of the fossils. All quantitative measures reported here were collected with Mitutoyo digital calipers. The cranial vault fragments were quantitatively compared to published data[66] in two analyses (Supplementary Note 2, Supplementary Fig. 5). First, the proportion of diploe relative to inner and outer bone layers for hominin parietal and frontal bones were compared via ternary plot comparison. Second, the absolute thickness of hominin frontal and parietal bones were compared among taxa by boxplot. The new fossils from Area 13 are housed at the NMK.

**Reporting summary.** Further information on research design is available in the Nature Research Reporting Summary linked to this article.

## Data availability

All data supporting the findings of this study are available within the paper, its supplementary information files, or as an upload to a data-sharing repository. Figure 3 source data in Google Earth format (.KMZ) are provided in Supplementary Data 1. Restrictions apply to the availability of hominin scan data figured herein, but these data are available from the corresponding author with the permission of the authorizing third party (museum or individual). Source data are provided with this paper.

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

## Acknowledgements

We thank the National Museums of Kenya, the Kenyan National Commission for Science, Innovation, and Technology (NACOSTI), the Kenyan Department of Mining, and the Daasenach community. Tim White, Christoph Zollikofer, Marcia Ponce de León, Bereket Haileab, Craig Feibel, Erin DiMaggio, Patrick Gathogo, Kyalo Manthi, Carrie Mongle, Ionuţ Şandric, and Louise Leakey are thanked for providing helpful conversations, data, and/or photographs. Purity Kiera, Robert Moru, Andrew Barr, Frances Forrest, Oumeyma Ben Brahim, Leah Myerholtz, Wendy Khumalo, Christopher Smith, and all other Koobi Fora Field School students and staff provided research support. Funding provided by the National Science Foundation (NSF REU 1852441, NSF 1358178, NSF 1624398), Fundação de Amparo à Pesquisa do Estado de São Paulo (FAPESP grant number 2018/208733-6), and the American Museum of Natural History.

## Author contributions

A.S.H. conceptualized the project. A.S.H., S.S.M., M.B., S.K., S.W., and D.V.P. collected data. A.S.H., S.S.M., Z.J., M.B., S.M., D.B.P., K.T.U., and D.V.P. analyzed data. A.S.H., S.S.M., D.R.B., E.N., D.B.P., K.T.U., and D.V.P. wrote the paper. All authors discussed the results and commented on the manuscript.

## Competing interests

The authors declare no competing interests.
