## [Peer Review File · Nature Communications]

Reviewers' Comments:

Reviewer #1:

Remarks to the Author:

I have added extensive comments in the attached MS. I think there are quite a few things that need to be explained or added to this study before it can be considered for publication. These include but are not limited to:

- 1) discrepancies in the naming of units between text and figures in the paper and supps.
- 2) issues with the way that the age of the site has been established including incorrect citation of papers to argue this case and how the erosional period in the sequence is dealt with
- 3) questions over the way that the composite stratigraphy has been put together from the various sections given that some layers have been omitted. Also questions over the location of those sections and how they relate to the hominin find sites
- 4) the structure of the paper and the repetitive nature of certain discussions.
- 5) the additional of more contextual photos, photogrammetry models that show the nature of the fossil occurrence itself rather than more general views of the basin or geology.
- 6) how the isotopic data is being used in the paper.
- 7) some photos of the faunal material to support the attributions as this material is not described at all and they should be in supps at least.

Reviewer #2:

Remarks to the Author:

The authors have relocated the East Turkana locality that provided the partial hominin occipital ER 2598. Based mainly on the presence of an occipital torus, this fossil has commonly been assigned to *Homo erectus*. In this paper and previous ones to which it refers, ER-2598 has been estimated to be 1.8-1.9 million years old. This would make it the oldest known fossil of *H. erectus*, supporting an African origin for the species. In their re-examination of the locality, the authors recovered a small number of non-hominin, mainly bovid fossils, some non-diagnostic hominin cranial fragments, a hominin proximal 3rd metatarsal and a hominin partial ilium. Both post-cranial fragments exhibit morphology compatible with *H. erectus*.

The antiquity of ER-2598 has been questioned because it was a surface find that might have been deflated from younger deposits. The authors observed no evidence of deflation in their fieldwork, but to support this conclusion, they should specify the process they think left the scattered bones including KNM ER 2598 and other fossils on the surface.

The manuscript is essentially a field report that does not advance knowledge of early human evolution, but it's clearly written and well-illustrated, and it will be useful to others who contemplate research at East Turkana. It merits publication.

Reviewer #3:

Remarks to the Author:

The authors do a nice job of clearly laying out the problem as well as the available solution; a recently identified fossil was previously misattributed to a collection area (15, ¹⁴Ma) but based on careful stratigraphic and sedimentological analysis, the research team supports their hypothesis that this fossil was, in fact, found in area 13 (early Pleistocene), at a stratigraphic horizon equal to ³m below the KBS tuff, surrounded by sandstones that resemble the UB1 sandstones (not the KB sandstones), and in the absence of evidence of deflation. The presence of other hominin fossils found nearby supports that the KNM-ER 2598 fossil is likely *Homo erectus*. Fossil and isotopic evidence of non-

hominins found in conjunction with the same horizon suggest a fluvially-influenced, open (grassy) environment.

A couple of quick comments and/or questions are below:

Supplemental info, Estimation of Locality Age Range- I'm not quite sure I follow the discussion of how the Upper Burgi member is a parasequence above the Borana Tuff (fluvial parasequences should fine upward, yes—is the context of this locality with respect to the Borana Tuff shown in Gathago and Brown, 2006?) and how that leads to an estimate of sedimentation rate and, therefore, age of the fossil(s). I see that the authors have cited Gathago and Brown (2006) and Feibel (1988), and perhaps clarity could be found in reading both of those papers. In the interest of clarity here, however, perhaps one sentence more in this section could help the reader make the link as to how the sedimentation rate was initially determined (maybe ranges for the upper Burgi type section from Feibel, 1988) and then why you chose the 52.4cm/kya rate specifically?

Line 99- "erosional window". Geologically speaking, an erosional window is when footwall rocks poke up through eroded hanging wall rocks. I don't think you're referring to any sort of fault feature here, so perhaps just using a different word than "erosional window" would be more appropriate and would avoid confusion with structural geologists?

Lines 114, 118, 120, 128- It appears in this section that you're describing the stratigraphy that is specifically depicted in Supp. Figure 2, section C (sections A and B don't show these same features). Could you please clarify this in this section (that you are describing section C, rather than a typical section in the area)?

Line 121- I'd be very suspicious if you truly found glauconite in your samples. Glauconite is a marine clay mineral (mainly authigenic but can be formed also by alteration of organic [fecal] material). It is easily identified by its greenish color, and should be easily identified by XRD (low angle 2-theta, to ID clay minerals). Also, do you truly have chlorite? Normally, chlorite is a metamorphic mineral (low grade), or (again) a marine authigenic clay mineral. Both chlorite and glauconite seem very unlikely given your reconstructed fluvial/terrestrial environment. A quick XRD analysis of the clay fraction should clarify both of these minerals. You may have misidentified chlorite and instead have a different phyllosilicate (muscovite? biotite?). With abundant potassium feldspar (microcline) and plagioclase in the region, I'm guessing you likely have kaolinite and/or montmorillonite (smectite) instead of glauconite. Montmorillonite and glauconite have similar 001 peaks in XRD.

Line 206- what is a lagas?

Lines 312-313 (and Figure 5)- Just curious, what do you speculate local δ_{w} ($\delta^{18}O$ of water, precipitation) at this time and elevation? That could give a good baseline value for your assertions of evaporative vs. humid.

Figure 3- What's your contour interval for the topo map in the background? The reason I ask this is b/c you mention in the text that you've scouted for fossils (and sandstone samples) within 50m of the KNM-ER 2598 site, and using the scale on the map, it appears that you could have changed elevation quite significantly within 50m of the site. Knowing the contour interval would put the reader's mind at ease that you're technically still in the same stratigraphic horizon(s) (generally speaking). (E.g., 50m to the west of KNM-ER 2598 site could get you to the same elevation as UB2, which is 2-3m stratigraphically below UB1, according to your strat column in Figure 3.).

Figure 4- These images are nice, but in my PDF they are small and it's hard to see what you describe in the caption (e.g., the aligned micas). I also hate that your ternary diagram (of the provenance of the sandstones) is in the supplemental info and not in the main text. This is just a suggestion, but... Since the images in Figure 4 are kindof small, perhaps choose 2 or 4 to show and then remove the

other 2 and add in the ternary diagram from Supplemental Info Figure 6. Supp Figure 6 really helps to solidify your conclusion that the KS and UB1 sandstones are unique, and I would encourage you to highlight that with your mineralogy discussion in Figure 4.

Figure 5- How did you quantify the environments (C3, Mixed, C4) on the x-axis of panel (c)? Whose data/model did you use? Cite it? Also, please fix the axes on panels (c) and (d) to read $\delta^{13}\text{C}\text{‰}$ (VPDB) and $\delta^{18}\text{O}\text{‰}$ (VPDB).

Cynthia M. Liutkus-Pierce

This revision incorporates the following:

- **Revised dates** based on alternate methodology. We removed the complex sedimentation model and now provide dates bracketed by the local tuffs, and have moved this section from SOM to the main text (per request of Reviewers 1 and 3).
- **Revised stratigraphic figure, moved from SOM to main text (new Fig2)**. The figure incorporates concerns raised by Reviewer 1 about how the erosional period is dealt with in the section, along with lateral correlations to two other published sections.
- **New Fig4** achieved by combining a supplemental and main text figure on sandstone mineralogy (as requested by Reviewer 3).
- **Revised Fig 6** shows additional views of comparative fossil pelvises with scales.
- **Extra Supplementary Figure (SFig2)** with views of the locality (as per Reviewer 1)
- **Extra Supplementary Figure (SFig7)** with additional views of contemporaneous pelvises from East Turkana, to support text added to Discussion.
- The original Fig4 was updated and moved to the SOM to become SFig1
- Reorganized Results text for better flow, with a resulting reordering of the figures.
- Revised Discussion, removing redundant text and elaborating on key details of the new hominin partial pelvis.
- Bibliographic references have updated throughout text.
- A new manuscript title.
- All minor suggested edits were incorporated.

All comments have been numbered to facilitate the responses below.

Reviewer 1

I have added extensive comments in the attached MS. I think there are quite a few things that need to be explained or added to this study before it can be considered for publication. These include but are not limited to:

- 1) discrepancies in the naming of units between text and figures in the paper and supps. *We have simplified our geology chapter to fit the purpose of this paper. During this simplification process, we have put extra attention in making sure the geologic terminology is uniform throughout the manuscript (Thank you for catching the error with the sandstone name in the KBS Member in the supplemental file! We regret this oversight, carried over from a prior version of the text).*
- 2) issues with the way that the age of the site has been established including incorrect citation of papers to argue this case and how the erosional period in the sequence is dealt with *The age model has been reformulated, focused on correlating geologic data from our area of investigation with existing geochronologic, lithologic and stratigraphic references. We would like to thank Reviewer 1 for their detailed comments which have helped us clarify these issues, and significantly improve the manuscript in the process. We address these issues in itemized responses below (reviewer comments numbered 15-26).*

3) questions over the way that the composite stratigraphy has been put together from the various sections given that some layers have been omitted. Also questions over the location of those sections and how they relate to the hominin find sites

Some of these issues relate to the prior comment, and we refer the reviewer to comments numbered 15-26 below. We acknowledge the discrepancies with the initial stratigraphy, which were due to the use of a broader set of sites and stratigraphic information, well beyond the scope of the paper. We have removed this excess data and focus our work in this paper on placing the sites discussed in the paper in the regional stratigraphic framework. We would like to draw the reviewers' attention to the revised stratigraphy figure (now main text Figure 2), which incorporates the corrections and diagrams where the sections derive relative to the hominin finds. We aim to further develop the discarded stratigraphic data in a standalone paper in the near future.

4) the structure of the paper and the repetitive nature of certain discussions.

Thank you. Although little can be done about the structure of the paper in terms of the Nature Communications organization, we have revised and reorganized in the Introduction, Results, and Discussion sections. We have specifically revised the Discussion to reduce the repetitive nature of the text by removing passages that reiterate the Results and narrow in on the implications. The largest change to the Discussion is that we have taken this helpful comment as an opportunity to elaborate on the implications of the new ilium for the Homo erectus hypodigm. Finally, we do summarize our findings in the final paragraph of the Discussion (which serves as a coda for the paper) and we now begin this paragraph with "In summary, ..." in order to make clear that this is a synthesis of the main findings.

5) the additional of more contextual photos, photogrammetry models that show the nature of the fossil occurrence itself rather than more general views of the basin or geology.

Additional photographic views of the location are provided in a new Supplementary Figure 2.

6) how the isotopic data is being used in the paper.

The enamel isotope data presented here add an important layer of ecological information to our understanding of this locality. Previous studies at East Turkana indicate that dietary habits within taxa (particularly grazers) can vary greatly across space (for example, Patterson et al. 2017, Journal of Human Evolution 112:148-161). Therefore, it is crucial that we provide direct evidence of diet in the form of isotopes, rather than rely on inferences based upon the taxonomic uniformitarianism, to understand the distribution of particular vegetation types on the ancient landscape.

7) some photos of the faunal material to support the attributions as this material is not described at all and they should be in supps at least.

We feel this suggestion is beyond the scope of this paper, for reasons addressed more fully below (comment 38).

[Itemized comments from PDF, with original lines indicated]

8) L1-2: the title could be a bit more descriptive of the outcomes of the paper, especially for a more general audience. This title will mean very little to people outside the discipline.

Thank you for the suggestion. We have selected a new title, "New hominin remains and revised context from the earliest Homo erectus locality in East Turkana, northern Kenya"

9) L38: this is an overly precise age given there has actually been no direct dating done at the site.

Agreed. We have tempered our language in regard to the narrow date here in the abstract and throughout the manuscript. The 1.88 Ma date was derived from a hypothetical sedimentation model, and the reviewer is correct to take issue with this narrow date (which has the potential to be misconstrued by readers). We have removed the sedimentation model and provide a date range that is bound by the overlying and underlying tuffs.

0) L48: ref?

Provided.

1) L60: The fossils have a maximum age of 1.78 Ma. This is the maximum age for the lithic assemblage, which is not the same thing if you are talking about the age of fossil remains of Homo erectus.

Thank you for noting this critical clarification. We have updated the text and our wording to be more specific (Lines 62-64).

2) L88: add height amsl.

Added.

3) L88: please provide a photo of this collapsed cairn both in close up and set in the landscape so the reconstructed location of the fossil can clearly be seen.

Additional views of the cairn and location are provided in a new supplemental figure (new Supplementary Figure 2). Other SOM figures have been renumbered and updated throughout the text.

4) L99-100: this description and the very weathered nature of the specimen is exactly why people are going to think that it might have been reworked from younger strata. Nowhere do you show the actual location of the KNM-ER 2598 or new fossils you have collected except in SF3C. That photo seems to suggest that the remnant of Burgi Member material is extremely eroded down to almost the same level at the drainage channel. You should include good photos showing the reconstructed location as well as preferably the location of the new fossils in situ. *Thank you for this comment. We have provided additional photos (new Supplementary Figure 2) which shows the flat topography of the KNM-ER 2598 reconstructed location and the areas nearby where fossils were recovered. The photos provide additional evidence that it is unlikely that the fossils were reworked from younger strata since the Burgi Member is eroded down to nearly the same level as the drainage (as noted by the reviewer).*

15) L118: In SF2 you seem to suggest it terminates with a conglomerate in Section B? You don't discuss the uncertainty suggested in SF2 with regards the association of the conglomerate to Burgi or KBS. Or is the pink line just showing the location of where the KBS tuff should be in Section C? the figure is unclear in this regard as the line seems to infer this is where you are putting the contact between Burgi and KBS. Moreover, if this is the case then why have you reconstructed the KBT as occurring under the conglomerate in Section C when Section B also contains a conglomerate with an basal disconformity below KBT in Section B. This is a critical issue as it would suggest that the fossils are lower within the Burgi Member than you have reconstructed. Why have you correlated that conglomerate with the one in the top of section A instead?

This section has been re-written to focus solely of the stratigraphic correlation between the area of interest and the regional stratigraphic framework. We clarified the correlations and corrected the text (see response 3).

16) L118: You further show no disconformity in Section A. So why are you defining the silt unit to Burgi?

Corrected in revisions.

17) L124: In Figure 3B you show the fossils as being above UB1 but in SF2 you show then as being below UB1. Which is it? another reason for good photos of the find spots to show this relationship.

Corrected in revisions. Photos also now provided in SOM.

18) L125-131: You call this a conglomerate in the figures so why a sandstone here? This still does not explain the meaning of your pink dotted line in Figure F2. You say the KBS Member Sandstone KS is thick with an erosive base, but so is the one in section B. How does that one differ to the KS?

Addressed. We have rewritten this section and the issues raised by the reviewer are no longer present in the text/figure.

19) L133: how have you defined this when you are seemingly uncertain (based on SF2) as to whether the KBT occurs above or below the KS sandstone in Section C based on the other sections. Surely you should give a range variation based on these uncertainties. Besides, i really don't see what difference it being 3m or 4m below KBS makes when there is a well defined erosional contact between the two Members. This makes any depth analysis like this a bit meaningless in terms of age calculation. It would also help to resolve the discrepancy showing the fossils below or above UB1 as this adds 1m of variation.

Addressed. We recognize that there were some issues with the stratigraphic section due to reliance on sections outside of our study area, and these need to be further investigated and published in a separate standalone paper. We have rewritten this section and the issues raised by the reviewer are no longer present in the text and figure (now main text Figure 2).

20) L135: reason a good photo or even better yet, photogrammetry model is needed.

Several additional photos are now provided in the SOM (Supp Fig 2) for better spatial context.

21) L137: where is this data? Do you need to reference it somewhere?

Citation to Figure 4 has been added.

22) L138: you actually call it [the sandstones] a 'paraconglomerate' in SF2 and conglomerate in the key of this figure and F3

Changed. The confusion is due to the lateral variations in the unit. The sandstone unit laterally grades into a conglomeritic sandstone. The revised manuscript describes this feature (Lines 128129).

23) L139-141: how does this help your argument? it means you have no idea what was on top of these deposits before they were eroded. They could have been KBS sediments from which the fossils have deflated, being surface finds. Did you do any form of excavation to show that fossils occurred in situ within the layers you are saying they come from?

The paragraph was misleading and has been rewritten/corrected. We focus on showing the continuous upper Burgi - KBS succession both in the manuscript text and in the figures (e.g., in the main text Figure 2 section A, and Supplementary Figure 4b).

24) L139, in reference to the "KNM-ER 2598 locality": how are you defining 'locality' here compared to your maps and clusters. It is shown in Section C of SF2. Where exactly are these sections A-B-C in SF2 when compared to the map in F3 for example if you are saying they are not present at the locality? When I compare Fig 3 and FS2 KBT is seen directly overlying clays of the Burgi formation and the fossil deposits. Of course this is shown to not be the case in SF2 where KBT does not occur in the same section as the fossils so that is obviously a composite section. However it seems to pick and chose what it shows in such a composite and I think this distorts the actual sequence of accumulation at the site:

First, thank you for mentioning this point about "locality." Properly defining the field research area has been a central challenge to the authors throughout the course of writing this paper. The words "site" and "collecting area" were discounted based on the potential for being inaccurate, and we adopted locality to refer to the two fossiliferous clusters in the Upper Burgi deposits near the KNM-ER 2598 reconstructed location. However, we see the potential for confusion in some locations in the manuscript, so we have revised the text to be more specific as to Cluster1 or Cluster2 wherever appropriate.

Second, in regard to the sections with features not present-- The figures have been corrected. We removed sections from outside of the area of study and composite sections have been split to better illustrate the location on the map.

We address Reviewer 1's numbered points individually below.

25) 1) it does not help that you call the laminated layers 'muds' in in SF2 and then 'clays' in F3. Please be consistent with your terminology between text and different Figures.

Corrected.

26) 2) section A and B in SF2 clearly show there is a conglomerate unit and a siltstone unit overlying these muds/clays and occurring below KBT, so why have these been omitted from the composite section in F3?

Corrected. See also response 15.

27) 3) these sections show that hominin fossils occur in two different sections A and C, but which hominin fossils. These should be named if they come from different sections. Again, knowing where these sections are on Figure 3 would help understand what is going on. I assume these are related to the 'geological observation points' but it is not clear how you have assembled these 10+m deep sections from this.

The figures have been corrected. We removed sections from outside of the area of study and composite sections have been split to better illustrate the location on the map. The relation between the fossil sites and their position in the stratigraphic figure (main text Figure 2) has been clarified.

28) L144: so from the surface then? No excavation?

There were no excavations performed, and we consider all of the fossil specimens to be surface finds, even the very few that were embedded in the surface sediment (for example, see Supplemental Figure 2f). We have updated the text to explicitly state that these were surface finds (Lines 254, 438-439, 133-134) and that all fossils were found sitting on the surface or partially embedded in sediment (Line 439).

29) L147: It is extremely common on such open landscapes for material to appear embedded within a unit but in fact it has been consolidated into slightly reworked sediments with a similar character. Some photos of this would add more evidence to such a claim, as of course would some excavations showing clear in-situ context.

This is a very good point. Upon revisiting our collection records, we only have a small number of fossils that were embedded in surface sediment (note that we never used the phrase in situ in this manuscript). Given the small number, and the point raised by the reviewer about reworked sediments giving misleading appearances of being in situ, we have clarified this in the collecting methods text. We do not want to give readers the appearance that we would characterize these finds as in situ (sensu stricto).

30) L190-191: reference to data supporting this vague statement 'consistent'? even if this is your Figure comparing them. because you then note how they are different to these other Homo specimens below.

This paragraph has been revised throughout to be more specific about the comparative iliac morphology of this specimen judged relative to other fossils. Edits have been made to the comparative figure (main text Figure 6) and an additional figure has also been provided (Supplementary Figure 7) in order to show the fossil morphology being discussed. These revisions have greatly benefitted the manuscript by allowing us to elaborate in the Discussion on the significance of the pelvis within the context of Homo erectus pelvic specimens.

31) L191: how?

See response to Comment #30 above.

32) L199: again, a vague statement. How is it similar?

See response to Comment #30 above.

33) L200: which means what?

See response to Comment #30 above.

34) L212: what age does the morphology of KNM-ER 2598 suggest exactly? it is a small fragment of likely *H. erectus* cranium, a species known to occur for 1.9 Mya. How does that suggest an expected age?

Rewritten. Thank you, we agree that this was confusing as previously written.

35) L224: As noted before, this seems debatable based on your sections. Moreover, there is a major erosional period between the two.

This section has been re-written to focus solely of the stratigraphic correlation between the area of interest and the regional stratigraphic framework. We show that in the study area the major erosional episode(s) occur(s) after the deposition of the KBS Tuff, in full agreement with the existing models. We acknowledge that there is a need to improve our stratigraphic models for the region. We initially presented sections outside of the study area, for a broader discussion but acknowledge that this is not the focus of the current article. Instead the broader stratigraphy and correlations issues will be addressed in a stratigraphy-focused paper, to be developed in the near future.

36) L227-228: this should be in the main text of the paper.

Thank you for this point, also raised to some extent by Reviewer 3. The dating section is now moved into the main text. Additionally, we have removed the complex sedimentation model entirely from this paper and instead constrain the age based on the bracketing tuffs. The minimum age is now limited by the overlying KBS Tuff, and the maximum age is limited by the underlying Borana Tuff. The literature regarding the Borana Tuff presents some challenges (namely, between-site correlations and interpretations of paleomagnetic polarity) that hamper a clear assignment of a lower age for the fossils discussed in the paper. We suggest an age constraint of >1.855 Ma based on fossil position below the KBS Tuff, and the maximum age would likely be around 1.934 Ma.

*I also have a bit of an issue with it: 1) an age of 1.88 Ma is actually within error of the age for KBS that you are using 1.876 +/- 0.021 Ma i.e 1.897-1.855 Ma. Given that there is a defined erosional contact between the Burgi and KBS Members then how are you simply assuming that there is virtually no loss of time between the age of the Burgi sediments and the KBS tuff? *We did not see an erosional contact in our study area, and the misleading correlation that led us to this interpretation has been removed from the manuscript. Additionally, the sedimentation model has been removed, so this is no longer a direct issue for the manuscript.**

2) You state that the lower age range relates to correlating the units to the position of the reversed polarity in the Borana Tuff in Area 10 and cite Gathogo and brown 2006 AES. That

paper is the formal establishment of the Borana Tuff but has no palaeomagnetic data or statement that it is reversed. So you need to clarify where this information is coming from.

We did extensive examination of the published works on the Borana Tuff magnetic polarity and stratigraphic correlations during this revision, and we now present this in the Discussion (Lines 265-281). We acknowledge in the text that there are limitations to interpreting the lower age constraint, and we further suggest additional targeted paleomagnetic and stratigraphic work is needed to clarify the constraining age provided by the Borana Tuff. We consulted with several external researchers who may have relevant unpublished information (Feibel, Cerling, Haileab, Gathogo, Grine) and are confident with the depth of the resolution we provide in this paper.

3) also see supp comments. *Addressed, here and in the Discussion.*

4) not sure how you calculate the sedimentation rate of 52.4cm/kya but it completely ignores the erosional contact and thus unknown period of time between the KBS tuff and the Burgi and hominin fossils. You would need to do your own palaeomagnetic analysis of the KNM-ER 2598 deposits to clarify an upper age limit. Not sure why you didn't do this, or at least show clear correlations to other dated sections to say anything more than it is older than the KBS tuff. *Sedimentation model has been removed (see prior comments). We suggest that paleomagnetic work needs to be done, in part because we are planning a larger scale paleomagnetic study of the region (including Area 10). We think that the published works provide enough data to develop maximum age constraints for the purposes of this paper.*

37) L239-247: this paragraph is a bit repetitive and could be condensed.

Paragraph rewritten and condensed.

38) L305-318: Why are there no photos or descriptions of the faunal material, at least in supps?

The structure of the paper is a bit odd as a lot of the discussion seems repetitive in terms of the geology and hominins where the results above includes much of the same sort of discussion but then the isotope date is just noted in the above results section and only then discussed here.

I'm also not sure what this data is bringing to the story of the context and location of this new material. You bring no new dates, which would be most critical but then add this data in. if this is an attempt to further show that the fossils come from Burgi based on their isotopes (as presented in STable 2 then to do this you can't just compare the values against Burgi fossils from elsewhere but also have to compare them against fossils from KBS where the argument is that the fossils could have come from.

The faunal material is used here to contextualize the KNM-ER 2598 site in terms of paleoecology and environmental conditions. We do not make biostratigraphic inferences based on the faunal material and, moreover, we only provide very generalized taxonomic resolution (usually not beyond the family or tribe level) which should not influence the isotopic data in any significant way.

While we appreciate the reviewer's request for additional documentation of the faunal material, faunal descriptions are typically reserved for large paleontological monographs. Fauna are frequently analyzed in this way (i.e., faunal abundance, isotopic data) prior to monograph

publication. It is not a common practice to extensively document the fauna except for papers that establish new species/genera etc., provide descriptive taxonomy, or deal with systematics.

If the reviewer is concerned about our identifications, we hope that they will feel some reassurance in our experience in identifying Pleistocene fauna to family/tribe level. Our authors have many years of experience making identification and more than 3 dozen papers demonstrating this skill.

As an additional note-- We have fully accessioned the faunal material, aside from a single relatively rare rhino specimen which will be studied more thoroughly at a future date (note that we do now provide a field photo of this rhino specimen in Supplemental Figure 2). Fully accessioning the fauna provides other researchers the opportunity to study the faunal material at the National Museums of Kenya. We believe we are promoting rapid open access to these fossils through this publication approach.

39) L319-336: again, this is very repetitive and repeats what you have just said above in the discussion

We provide this summarizing paragraph because Nature Communications does not include a Conclusion section and only allows for very abbreviated abstracts, making this an important summary of the manuscript. To make it more clear that this is the coda, we begin the paragraph with "In summary,..."

40) SOM page 2: You state that the lower age range relates to correlating the units to the position of the reversed polarity in the Borana Tuff in Area 10 and cite Gathogo and brown 2006 in AES. That paper is the formal establishment of the Borana Tuff but has no palaeomagnetic data or statement that it is reversed. So you need to clarify where this information is coming from

See comment 36 above. We have elaborated on this in the Discussion.

41) SOM page 2: based on what? where is the data for this?

Removed. The sedimentation model has been removed.

42) how is this sedimentation rate calculated? It also ignores the erosional period occurring between the dated tuff and the Burgi Member

Reformulated/removed. See comment 36.

Reviewer 2

The authors have relocated the East Turkana locality that provided the partial hominin occipital ER 2598. Based mainly on the presence of an occipital torus, this fossil has commonly been assigned to *Homo erectus*. In this paper and previous ones to which it refers, ER-2598 has been estimated to be 1.8-1.9 million years old. This would make it the oldest known fossil of *H. erectus*, supporting an African origin for the species. In their re-examination of the locality, the authors recovered a small number of non-hominin, mainly bovid fossils, some non-diagnostic

hominin cranial fragments, a hominin proximal 3rd metatarsal and a hominin partial ilium. Both post-cranial fragments exhibit morphology compatible with *H. erectus*.

43) The antiquity of ER-2598 has been questioned because it was a surface find that might have been deflated from younger deposits. The authors observed no evidence of deflation in their fieldwork, but to support this conclusion, they should specify the process they think left the scattered bones including KNM ER 2598 and other fossils on the surface.

Incorporated. We have now specified that the most likely interpretation of all of the data gathered is that the fossils have weathered out of the UB1 sandstones. We do not fully exclude deflation processes but we show major alluvial components in washing and eroding the surface. This means that deflation might play a role in the unearthing of the fossils but constant wash and transport during the rain episodes washes these remains from the site (as it washed away material from the KBS).

The manuscript is essentially a field report that does not advance knowledge of early human evolution, but it's clearly written and well-illustrated, and it will be useful to others who contemplate research at East Turkana. It merits publication.

Reviewer 3

The authors do a nice job of clearly laying out the problem as well as the available solution; a recently identified fossil was previously misattributed to a collection area (15, ~4Ma) but based on careful stratigraphic and sedimentological analysis, the research team supports their hypothesis that this fossil was, in fact, found in area 13 (early Pleistocene), at a stratigraphic horizon equal to ~3m below the KBS tuff, surrounded by sandstones that resemble the UB1 sandstones (not the KB sandstones), and in the absence of evidence of deflation. The presence of other hominin fossils found nearby supports that the KNM-ER 2598 fossil is likely *Homo erectus*. Fossil and isotopic evidence of non-hominins found in conjunction with the same horizon suggest a fluvially-influenced, open (grassy) environment.

A couple of quick comments and/or questions are below:

44) Supplemental info, Estimation of Locality Age Range- I'm not quite sure I follow the discussion of how the Upper Burgi member is a parasequence above the Borana Tuff (fluvial parasequences should fine upward, yes—is the context of this locality with respect to the Borana Tuff shown in Gathago and Brown, 2006?) and how that leads to an estimate of sedimentation rate and, therefore, age of the fossil(s). I see that the authors have cited Gathago and Brown (2006) and Feibel (1988), and perhaps clarity could be found in reading both of those papers. In the interest of clarity here, however, perhaps one sentence more in this section could help the reader make the link as to how the sedimentation rate was initially determined (maybe ranges for the upper Burgi type section from Feibel, 1988) and then why you chose the 52.4cm/kya rate specifically?

Reformulated/removed. See comment 36.

45) Line 99- “erosional window”. Geologically speaking, an erosional window is when footwall rocks poke up through eroded hanging wall rocks. I don’t think you’re referring to any sort of fault feature here, so perhaps just using a different word than “erosional window” would be more appropriate and would avoid confusion with structural geologists?

Corrected.

46) Lines 114, 118, 120, 128- It appears in this section that you’re describing the stratigraphy that is specifically depicted in Supp. Figure 2, section C (sections A and B don’t show these same features). Could you please clarify this in this section (that you are describing section C, rather than a typical section in the area)?

Corrected.

47) Line 121- I’d be very suspicious if you truly found glauconite in your samples. Glauconite is a marine clay mineral (mainly authigenic but can be formed also by alteration of organic [fecal] material). It is easily identified by its greenish color, and should be easily identified by XRD (low angle 2-theta, to ID clay minerals). Also, do you truly have chlorite? Normally, chlorite is a metamorphic mineral (low grade), or (again) a marine authigenic clay mineral. Both chlorite and glauconite seem very unlikely given your reconstructed fluvial/terrestrial environment. A quick XRD analysis of the clay fraction should clarify both of these minerals. You may have misidentified chlorite and instead have a different phyllosilicate (muscovite? biotite?). With abundant potassium feldspar (microcline) and plagioclase in the region, I’m guessing you likely have kaolinite and/or montmorillonite (smectite) instead of glauconite. Montmorillonite and glauconite have similar 001 peaks in XRD.

Removed. Petrographic descriptions no longer make reference to glauconite and chlorite, as this does not affect the interpretation in this study. The presence or absence of these minerals will be tested and presented elsewhere.

48) Line 206- what is a lagas?

Changed. Lagas (=ephemeral stream in the local Gabra language) has been replaced with ephemeral streams.

49) Lines 312-313 (and Figure 5)- Just curious, what do you speculate local $\delta^{18}O$ of water, precipitation) at this time and elevation? That could give a good baseline value for your assertions of evaporative vs. humid.

Thank you. Although we would love to speculate more regarding localized water composition, we feel that it would be preliminary at this time given our comparatively small sample size from this locality. It will certainly be something that we investigate in the future. To make this even more clear, we have replaced "suggests" with "potentially indicates" (line 354) more arid local conditions. We feel that this level of speculation is appropriate given the overall pattern of $\delta^{18}O$ enrichment (relative to existing UB data) in our data as noted in Figure 5d.

50) Figure 3- What’s your contour interval for the topo map in the background? The reason I ask this is b/c you mention in the text that you’ve scouted for fossils (and sandstone samples) within 50m of the KNM-ER 2598 site, and using the scale on the map, it appears that you could

have changed elevation quite significantly within 50m of the site. Knowing the contour interval would put the reader's mind at ease that you're technically still in the same stratigraphic horizon(s) (generally speaking). (E.g., 50m to the west of KNM-ER 2598 site could get you to the same elevation as UB2, which is 2-3m stratigraphically below UB1, according to your strat column in Figure 3.).

To best show the terrain in the figure we used a space of 0.5m between the contour lines. We have also clarified this in the figure caption to avoid misleading interpretations of topography (which is actually quite flat around the KNM-ER 2598 site).

51) Figure 4- These images are nice, but in my PDF they are small and it's hard to see what you describe in the caption (e.g., the aligned micras). I also hate that your ternary diagram (of the provenance of the sandstones) is in the supplemental info and not in the main text. This is just a suggestion, but.... Since the images in Figure 4 are kindof small, perhaps choose 2 or 4 to show and then remove the other 2 and add in the ternary diagram from Supplemental Info Figure 6. Supp Figure 6 really helps to solidify your conclusion that the KS and UB1 sandstones are unique, and I would encourage you to highlight that with your mineralogy discussion in Figure 4.

This is an excellent suggestion and we have followed the reviewer's advice. Specifically, we have reduced the number of thin section images in Figure 4 and added the triangle plot from Supplemental Fig 6. The revised figure provides much more information for the reader and we thank the reviewer again for this helpful comment.

52) Figure 5- How did you quantify the environments (C3, Mixed, C4) on the x-axis of panel (c)? Whose data/model did you use? Cite it? Also, please fix the axes on panels (c) and (d) to read $¹³C\%$ (VPDB) and $¹⁸O\%$ (VPDB).

Thank you. We have added text describing our method for calculating %C4 from the tooth enamel data (Lines 448-452). Figure axis has been updated.

Reviewers' Comments:

Reviewer #1:

Remarks to the Author:

I think the authors have done a great job replying to my comments and making the paper easier to follow and the argument coherent. It is an interesting addition to the history of an important fossil, as well as a nice addition in terms of new postcranial hominin material and it should ultimately be published.

However, I have one remaining issue with their reasoning for not being able to set an upper age limit for the fossils based on a review of their argument and the papers they cite. I have made comments on the MS itself to show what I mean but here is the main issue I have based on the references provided:

Chapon et al 2011 show the Borana Tuff at Koobi Fora occurring above the Lorenyang Tuff dated to $^{147}\text{Sm}/^{143}\text{Nd}$ 1.90 Ma while there is an error of 0.05 Ma on this age that suggests Borana is <1.95 Ma and you are saying your fossils are younger than the Borana tuff so even younger.

Moreover, Chapon also state (page 257) that this Tuff in Upper G at Shungura is the 'G29' Tuff which Kidane et al 2007 show is within a normal polarity and thus within Olduvai. So using your own citations and argument your fossils cannot be older than the base of the Olduvai as the Borana tuff is within the base of Olduvai and your fossils are younger than it.

thus throughout the paper you could actually give it an upper age range of 1.934 Ma or 1.95 Ma (depending which date you wish to use as the correct age is still a matter of debate) based on your argument.

I have a few other minor comments in the MS, although a critical one, as it is incorrect, is that the MS states the 2.04 Ma age adjacent to DNH 134 from Drimolen is a uranium-lead age when actually it is an ESR age. U-Pb dated the flowstone overlying the fossils to 1.96 Ma, at the same time the base of the Olduvai SubChron is recorded.

Reviewer #3:

Remarks to the Author:

This revised manuscript does a nice job of clearly laying out the problem as well as the available solution; a recently identified fossil was previously misattributed to a collection area (15, $^{147}\text{Sm}/^{143}\text{Nd}$ 4Ma) but based on careful stratigraphic and sedimentological analysis, the research team supports their hypothesis that this fossil was, in fact, found in area 13 (early Pleistocene), at a stratigraphic horizon equal to $^{147}\text{Sm}/^{143}\text{Nd}$ 3m below the KBS tuff, surrounded by sandstones that resemble the UB1 sandstones (not the KB sandstones), and in the absence of evidence of deflation. The presence of other postcranial hominin fossils found nearby could represent the oldest *H. erectus* postcrania, and support the assertion that the KMN-ER 2598 fossil is likely *Homo erectus*. Fossil and isotopic evidence of non-hominins found in conjunction with the same horizon suggest a fluvially-influenced, open (grassy) yet arid environment.

The reviewers provided an incredible amount of feedback to the authors on this manuscript, so with that said, the authors had a lot of information to deal with; and I feel they did an excellent job revising this manuscript taking into account the copious suggestions/comments. This revised manuscript reads very smoothly and I am impressed by its fluidity. All of my questions have been addressed by either removing the questionable information or clarifying the information in the text. I think Figure 4 is much more impactful now as well, and really stresses to the reader the differences/uniqueness in the sandstone, which is a major component of the authors' assertion of "no deflation". In short, this is an excellent revision. Not only were all of my comments addressed

thoroughly, but the authors also spent considerable time addressing the numerous suggestions of the other reviewers. By doing so, the paper is, in my opinion, ready for publication with just a couple of tiny grammatical corrections (noted in purple on the attached PDF). I'm sure these would be fixed during the editing process, so I do not feel that this manuscript needs to go through further revisions. Nice job, and interesting work.

Reviewer #1 (Remarks to the Author)

I think the authors have done a great job replying to my comments and making the paper easier to follow and the argument coherent. It is an interesting addition to the history of an important fossil, as well as a nice addition in terms of new postcranial hominin material and it should ultimately be published. However, I have one remaining issue with their reasoning for not being able to set an upper age limit for the fossils based on a review of their argument and the papers they cite. I have made comments on the MS itself to show what I mean but here is the main issue I have based on the references provided:

Thank you again to Reviewer 1. We are very happy with the changes that we made to the manuscript in response to the constructive points raised.

Chapon et al 2011 show the Borana Tuff at Koobi Fora occurring above the Lorenyang Tuff dated to ~1.90 Ma while there is an error of 0.05 Ma on this age that suggests Borana is <1.95 Ma and you are saying your fossils are younger than the Borana tuff so even younger.

Moreover, Chapon also state (page 257) that this Tuff in Upper G at Shungura is the 'G29' Tuff which Kidane et al 2007 show is within a normal polarity and thus within Olduvai. So using your own citations and argument your fossils cannot be older than the base of the Olduvai as the Borana tuff is within the base of olduvai and your fossils are younger than it.

thus throughout the paper you could actually give it an upper age range of 1.934 Ma or 1.95 Ma (depending which date you wish to use as the correct age is still a matter of debate) based on your argument.

Thank you for noting that we approached some correlations in a conservative way. Our intention was not to solve the stratigraphic issues so we kept the discussion in the manuscript to a necessary minimum. But we will explain more here to clarify why the lower age constraint based on the Borana Tuff is “blurry”, and why a lower age constraint based on the Lorenyang Tuff would be entirely problematic.

*In the case of the Borana correlations of Chapon et. al. (2011), we have already followed Reviewer1’s advice (see our manuscript lines 292-293). Chapon et al. chose to laterally correlate the Borana Tuff simply to some portion of the upper G Member of the Shungura Formation. Specifically, Chapon et al. (2011) write: “On the basis of their geochemical composition, the **Borana Tuff may be correlated with the G-29 Tuff located toward the top of the Upper G member** (Haileab and Brown, 1994). **As this latter tuff was described outside the Upper G member type section, it is preferable to mention that the Borana Tuff is correlated with an unnamed tuff from the Upper G member in contrast to our previous proposal (Chapon et al., 2008).”***

Thus, Chapon et al. acknowledge uncertainty in a specific correlation (even contradicting their past published work). We follow their lead, writing:

*“This discontinuous tuff occurs within a complex stratigraphic succession and past correlations resulted in conflicting, often contradictory, stratigraphic positions^{31,32} and association with both normal and reverse paleomagnetic polarity intervals^{17,33-35}. **A conservative approach³¹ correlated the Borana Tuff with “an unnamed tuff in the upper G Member” of the Shungura Formation.** This assignment does not exclude a reverse polarity (C2r.1r, beginning of the Olduvai Subchron) for the tuff³⁵ that would expand the range of the UB1.” Lines 290-293.*

Reviewer1 also suggested to use Lorenyang tuff in the age calculation. Indeed, we have investigated the possibility of using this tuff for our age model but our study revealed that at this stage the tuff is not a reliable stratigraphic marker for several reasons. It is correct that Chapon et. al. (2011) plot (in their Figure 3) a synthetic log of the Koobi Fora Formation showing the Lorenyang Tuff below the Borana Tuff, but this information has not been confirmed by observations in the Koobi Fora region. The Lorenyang tuff outcrops only in very few locations (Areas 102, 104 only according to Haileab, 1995) and has been confirmed to have a normal magnetic polarity (Joordens et al. 2011), but at this stage there is no known direct relationship between it and the Borana Tuff. It is thus a major uncertainty on which of these two tuffs is older. Less important but worth mentioning, the Lorenyang Tuff age (1.90 ±0.05 Ma) is also not a reliable radiometric age, but an estimation, “derived from linear interpolation between the base of the Olduvai Subchron and the KBS Tuff” (Joordens et al. 2011). Therefore, we believe that the Lorenyang

Tuff has little to no stratigraphic significance for the context of our paper and using this tuff would be very speculative and potentially detrimental in the long run.

What is agreed among different authors is that the Borana Tuff correlates to the upper G Member of the Shungura, probably ~1.934 Ma but not precluding >1.95 Ma (later date based on Grine et al., 2019 J Hum Evolution). We have written out text in such a way that we show preference for the younger date but we do not (and cannot) exclude the older date. This paper focuses on our fieldwork in Area 13 and we must accept that the stratigraphy of Koobi Fora cannot be resolved in this paper.

I have a few other minor comments in the MS, although a critical one, as it is incorrect, is that the MS states the 2.04 Ma age adjacent to DNH 134 from Drimolen is a uranium-lead age when actually it is an ESR age. U-Pb dated the flowstone overlying the fossils to 1.96 Ma, at the same time the base of the Olduvai SubChron is recorded.

Corrected. We have rectified the age dating method reference. Indeed, as Reviewer1 pointed, several methods have been used in Drimolen and ESR has provided the 2.04Ma age.

Reviewer #1 (Detailed comments from PDF)

(Introduction, p. 4) The 2.04 Ma age is a uranium series electron spin resonance age, not a uranium-lead age. The fossil is below a flowstone dated to ~1.96 Ma based on uranium lead dating and the identification of the reversal at the base of the Olduvai SubChron.

Corrected.

(Introduction, p. 4) It is actually C2r.1r.

Thank you for noting the typographical error. We have updated to C2r.1r.

The jury is also still out on whether the Olduvai base is this young. See Rivera et al 2017 as one example where they show normal polarity in lava flows dated to 1.948 Ma. Also reviewed in Channell et al 2020 QSREPSL. This is not big deal in the scheme of this paper but the authors seem to be trying to reinterpret the age of the DNH 134 specimen rather than simply citing its published age.

We understand Reviewer1's concerns. We use the most recent Geomagnetic Polarity Time Scale throughout the paper for consistency. Our paper is not aiming to reinterpret the age of the DNH 134 specimen, but we cannot use the 2020 geomagnetic standards for eastern Africa and use outdated geomagnetic standards for Drimolen in the same paper, particularly as it relates to a discussion of a single subchron (C2r.1r). While we understand the Reviewer has concerns about the age of the Olduvai base, that discussion is beyond the scope of this paper and ultimately to be decided by the International Commission on Stratigraphy and the International Union of Geological Sciences.

We have rewritten the sentence with the intention of being accurate but not directly suggesting an alternate date for Drimolen. The sentence now reads "The most recent Geomagnetic Polarity Time Scale (GPTS 2020) associate the C2r.1r chron to the 1.934-2.120 Ma time interval^{9,10}." We hope this is more acceptable.

(Introduction, p.4) They are dated to sometime after 1.78 Ma as they occur in reversed polarity post the Olduvai Subchron but it is assumed it is soon after.

Rewritten sentence per this suggestion.

(Discussion, p. 13) do you not want to state the age here just to remind readers.

Added

(Discussion, p. 14) C2r.1r is not the beginning of the Olduvai SubChron or do you mean up until the base of the Olduvai SubChron? Note that SubChron is the term used for Olduvai now as it is a well established long period of polarity, not event see Channell et al 2020 or Singer 2014 or indeed Ogg 2020

Changed from "event" to "Subchron". We have corrected the unfortunate choice of words that could have misled reader to believe we affirm the Olduvai Subchron is a magnetic excursion.

(Discussion, p. 14) Chapon et al 2011 clearly show the Borana Tuff at Koobi Fora occurring above the Lorenyang Tuff dated to ~1.90 Ma so how does this argument hold true? while there is an error of 0.05 Ma on this age that suggests Borana is <1.95 Ma and you are saying your fossils are younger than the Borana tuff so even younger.

Addressed previously (see above). The relationship between the Lorenyang Tuff and Borana Tuff is not actually known.

Moreover, Chapon also state (page 257) that this Tuff in Upper G at Shungura is the G29 Tuff which Kidane et al 2007 show is within a normal polarity and thus within Olduvai. So using your own citations and argument your fossils cannot be older than the base of the Olduvai as the Borana tuff is within the base of olduvai and your fossils are younger than it. thus throughout the paper you could actually give it an upper age range of 1.934 Ma or 1.95 Ma (depending which date you wish to use) based on your argument.

Addressed previously (see above). Both the lateral correlation to the Shungura and the geomagnetic polarity of the Borana Tuff are unknown and/or unresolved.

Reviewer #3 (Remarks to the Author)

This revised manuscript does a nice job of clearly laying out the problem as well as the available solution; a recently identified fossil was previously misattributed to a collection area (15, ~4Ma) but based on careful stratigraphic and sedimentological analysis, the research team supports their hypothesis that this fossil was, in fact, found in area 13 (early Pleistocene), at a stratigraphic horizon equal to ~3m below the KBS tuff, surrounded by sandstones that resemble the UB1 sandstones (not the KB sandstones), and in the absence of evidence of deflation. The presence of other postcranial hominin fossils found nearby could be represent the oldest *H. erectus* postcrania, and support the assertion that the KNM-ER 2598 fossil is likely *Homo erectus*. Fossil and isotopic evidence of non-hominins found in conjunction with the same horizon suggest a fluvially-influenced, open (grassy) yet arid environment.

The reviewers provided an incredible amount of feedback to the authors on this manuscript, so with that said, the authors had a lot of information to deal with; and I feel they did an excellent job revising this manuscript taking into account the copious suggestions/comments. This revised manuscript reads very smoothly and I am impressed by its fluidity. All of my questions have been addressed by either removing the questionable information or clarifying the information in the text. I think Figure 4 is much more impactful now as well, and really stresses to the reader the differences/uniqueness in the sandstone, which is a major component of the authors assertion of "no deflation". In short, this as an excellent revision. Not only were all of my comments addressed thoroughly, but the authors also spent considerable time addressing the numerous suggestions of the other reviewers. By doing so, the paper is, in my opinion, ready for publication with just a couple of tiny grammatical corrections (noted in purple on the attached PDF). I'm sure these would be fixed during the editing process, so I do not feel that this manuscript needs to go through further revisions. Nice job, and interesting work.

All minor typographical edits and suggestions have been incorporated. Thank you for your very helpful reviews. We are very pleased with the suggestions that you offered and feel the manuscript is much stronger after both round of revisions.